# A chemosensory-like histidine kinase is dispensable for chemotaxis *in vitro* but regulates the virulence of *Borrelia burgdorferi* through modulating the stability of RpoS

Ching Wooen Sze[1], Kai Zhang[1], Michael J. Lynch[2], Radha Iyer[3], Brian R. Crane[2], Ira Schwartz[3], Chunhao Li[1,4]*

1 Department of Oral Craniofacial Molecular Biology, Virginia Commonwealth University, Richmond, Virginia, United States of America, 2 Department of Chemistry and Chemical Biology, Cornell University, Ithaca, New York, United States of America, 3 Department of Pathology, Microbiology, and Immunology, New York Medical College, Valhalla, New York, United States of America, 4 Department of Microbiology and Immunology, Virginia Commonwealth University, Richmond, Virginia, United States of America

* cli5@vcu.edu.

**Data Availability Statement:** The authors confirm that all data underlying the findings are fully available without restriction. All relevant data are

## Abstract

As an enzootic pathogen, the Lyme disease bacterium *Borrelia burgdorferi* possesses multiple copies of chemotaxis proteins, including two chemotaxis histidine kinases (CHK), CheA$_1$ and CheA$_2$. Our previous study showed that CheA$_2$ is a genuine CHK that is required for chemotaxis; however, the role of CheA$_1$ remains mysterious. This report first compares the structural features that differentiate CheA$_1$ and CheA$_2$ and then provides evidence to show that CheA$_1$ is an atypical CHK that controls the virulence of *B. burgdorferi* through modulating the stability of RpoS, a key transcriptional regulator of the spirochete. First, microscopic analyses using green-fluorescence-protein (GFP) tags reveal that CheA$_1$ has a unique and dynamic cellular localization. Second, loss-of-function studies indicate that CheA$_1$ is not required for chemotaxis *in vitro* despite sharing a high sequence and structural similarity to its counterparts from other bacteria. Third, mouse infection studies using needle inoculations show that a deletion mutant of CheA$_1$ (*cheA$_1$$^{mut}$*) is able to establish systemic infection in immune-deficient mice but fails to do so in immune-competent mice albeit the mutant can survive at the inoculation site for up to 28 days. Tick and mouse infection studies further demonstrate that CheA$_1$ is dispensable for tick colonization and acquisition but essential for tick transmission. Lastly, mechanistic studies combining immunoblotting, protein turnover, mutagenesis, and RNA-seq analyses reveal that depletion of CheA$_1$ affects RpoS stability, leading to reduced expression of several RpoS-regulated virulence factors (i.e., OspC, BBK32, and DbpA), likely due to dysregulated *clpX* and *lon* protease expression. Bulk RNA-seq analysis of infected mouse skin tissues further show that *cheA$_1$$^{mut}$* fails to elicit mouse *tnf-α*, *il-10*, *il-1β*, and *ccl2* expression, four important cytokines for Lyme disease development and *B. burgdorferi* transmigration. Collectively, these results reveal a unique role and regulatory mechanism of CheA$_1$ in modulating virulence factor expression and add new insights into understanding the regulatory network of *B. burgdorferi*.

within the paper and its Supporting Information files.

**Funding:** This project is supported by following NIH grants: AI078958 to C.L; AI148844 to B.R.C and C.L; R35122535 to B.R.C; and AI045801 to I.S. The funders had no role in study design, data collection and analysis, decision to publish, or preparation of the manuscript. There are no authors received salary from the funder.

**Competing interests:** The authors have declared that no competing interests exist.

## Author summary

Lyme disease is an infectious disease caused by the bacterium *Borrelia burgdorferi* which is transmitted to humans through the bite of infected blacklegged ticks. It is the most commonly reported tick-borne illness in North America and Europe. *B. burgdorferi* is a highly invasive bacterium that can swim and penetrate deep into tissues to cause a wide range of disorders including arthritis, meningitis, and migratory musculoskeletal pain. This invasiveness is driven by the ability of the spirochete to sense and respond to its surrounding initiated by signal transduction coupled with a sophisticated array of sensory proteins located at both cell ends. The genome of *B. burgdorferi* encodes multiple copies of chemotaxis-related genes and the function of many of these genes remains unknown. In this report, we focus on deciphering the role of one of the chemotaxis histidine kinases, $CheA_1$. Our results reveal a novel regulatory contribution of $CheA_1$ to the infectious life cycle and pathogenesis of *B. burgdorferi*.

## Introduction

Chemotaxis allows motile bacteria to sense surrounding environments and accordingly navigate their swimming behaviors to promote their survival and ability to colonize and invade a host [1]. The bacterial chemotaxis signaling pathway relies on a two-component system (TCS) that contains a chemotaxis histidine kinase CheA and a response regulator CheY. In *Escherichia coli*, the histidine kinase CheA is a key component of the chemotaxis signaling pathway [1–4]. Chemotaxis is initiated when the membrane bound methyl-accepting chemotaxis proteins (MCPs) bind a ligand (i.e., attractants) leading to activation of signal transduction from the MCP to the histidine kinase CheA via a linker protein, CheW. Activated CheA through autophosphorylation then transfers its phosphate group to the chemotaxis response regulator CheY, which in turn binds to the flagellar motor switch proteins to modulate flagellar motor rotation allowing cells to swim toward or away from an environment.

Lyme disease (LD), also known as Lyme borreliosis, is the most commonly reported tick-borne illness in the United States and Europe, with more than 476,000 new cases annually in the U.S. [5]. LD is caused by the bacterium *Borrelia burgdorferi* (also known as *Borreliella burgdorferi*) which is transmitted to humans through the bite of infected *Ixodes* ticks. Upon dissemination from the bite site, highly motile spirochetes can invade other tissues and distal parts of the body. Early-stage LD can be treated effectively with antibiotics. Unrecognized LD, however, can have debilitating long-term effects, such as arthritis, meningitis, and migratory musculoskeletal pain, which can appear after months or even years of infection [6,7].

The genome of *B. burgdorferi* encodes for two *cheA* (*cheA₁* and *cheA₂*), three *cheY* (*cheY₁*, *cheY₂*, and *cheY₃*), and three *cheW* (*cheW₁*, *cheW₂*, and *cheW₃*) [8,9]. The majority of these genes are located within two gene clusters: *cheA₂-cheW₃-cheX-cheY₃*, and *cheW₂-bb0566-cheA₁-cheB₂-bb0569-cheY₂* [9–11]. The current evidence has revealed that all of the chemotaxis genes in the *cheA₂* cluster are required for chemotaxis [11–13]. In contrast, the genes in the *cheW₂* cluster that have been studied (with the exception of *bb0569*, which encodes for an atypical MCP [14]) are not involved in chemotaxis *in vitro*. For instance, *cheW₂* and *cheY₂* mutants have a similar *in vitro* phenotype as the wild type with no apparent defect in chemotactic function [11,13,15,16]. Our previous study showed that $CheA_2$ was essential for the chemotaxis of *B. burgdorferi*, e.g., the *cheA₂* mutant failed to reverse, ran constantly in one direction, and was non-chemotactic to attractants [11]. In addition, *in vivo* studies showed

that CheA$_2$ was essential for *B. burgdorferi* to establish infection in a mammalian host as well as to accomplish its enzootic life cycle [17].

The role of CheA$_1$ currently remains unknown. CheA$_1$ contains all the characteristic domains of a histidine kinase similar to CheA$_2$ but the P2 domain required for the binding of CheY response regulator is not as conserved as other known CheAs [11]. A recent *in vivo* study by Xu *et al.*, as well as our group, indicated that *cheY$_2$* and *cheW$_2$* are dispensable for chemotaxis and motility of *B. burgdorferi in vitro* but essential for a productive infection within the vertebrate host [15,18]. In order to comprehend the function of the *cheW$_2$* operon, and in particular CheA$_1$, in the chemotaxis and pathogenicity of *B. burgdorferi*, a *cheA$_1$* mutant was generated in the virulent B31 A3-68 Δ*bbe02* background [19] and its phenotype was characterized in great detail using different *in vitro* and *in vivo* approaches. The ability of this mutant to complete the enzootic infection cycle was analyzed and its contribution to RpoS-mediated regulation was explored.

## Results

### Structural comparison of *B. burgdorferi* CheA$_1$ and CheA$_2$

To investigate why CheA$_1$ is dispensable for chemotaxis *in vitro* while CheA$_2$ is required [11], we generated AlphaFold [20] models of the two CheA proteins (BbCheA$_1$ and BbCheA$_2$) and compared them to the solved structure of CheA from *Thermotoga maritima* (Tm) [21,22] (**Fig 1A**). The models show high confidence (pLDDT scores >80) for all domains in each monomer, with pLDDT scores < 50 only for the residues that comprise the P1-P2 and P2-P3 linkers. The domain composition of TmCheA is similar to BbCheA$_1$ and BbCheA$_2$ with some notable differences (**Fig 1B**). BbCheA$_1$ has an analogous domain arrangement compared to TmCheA with the only major difference being an extended P2-P3 linker (73 residues-long compared to 42 residues of TmCheA). In contrast, BbCheA$_2$ differs from BbCheA$_1$ and TmCheA in several distinct ways. First, BbCheA$_2$ has two P2 domains (CheY binding domains, referred to here as P2$^\alpha$ and P2$^\beta$) compared to a single P2 domain in TmCheA and BbCheA$_1$. Superimposition of the P2$^\alpha$ and P2$^\beta$ domains reveals a high degree of structural similarity despite a mere 26% identity (**Fig 1C**). Sequence alignment suggests that both P2 domains bind to CheY, which could potentially act to increase the local phospho-CheY concentration within the chemosensory arrays of *B. burgdorferi* (**S1A Fig**), as has been shown for the P2 domain of the canonical *E. coli* CheA [23]. The second major difference between BbCheA$_1$ and BbCheA$_2$ is that BbCheA$_2$ has an extended P3 domain (**Fig 1B**). CheA homodimerizes via P3-P3' interactions, wherein two adjacent P3 domains come together to form a four-helix bundle [21]. In BbCheA$_2$, the 122 residues in P3 domain is significantly longer than the ~53 residues domain found in TmCheA and BbCheA$_1$ due to inserts between residues Ile446-Leu515 (**Fig 1A and 1B**).

To gain more insight into the overall differences in the domain architecture of CheA$_1$ and CheA$_2$, we superimposed domains P1-P5 from BbCheA$_1$ (pink) and BbCheA$_2$ (green) to the analogous domains of TmCheA (tan, **Fig 1D**). Overall, the P1, P4, and P5 domains are quite similar in all three proteins (RMSDs < 1), whereas the analogous P2 and P3 domains vary substantially (RMSD ranging 0.71–10.2 Å) among the three CheAs. For the P2 domain superimposition, the domain cores are similar with most of the structural deviations arising from the loops connecting the β-strands, and the N- and C-termini connecting to the P1-P2 and P2-P3 linkers, respectively. Analysis of the putative CheY binding site for BbCheA$_1$ P2 and BbCheA$_2$ P2 (α and β) reveals that the structures of the P2 domains are generally well-conserved in this region, supporting sequence alignment data that indicates conservation in the CheY:P2 binding interface residues (**S1A–S1B Fig**). Superimposition of the P3 domains from TmCheA, BbCheA$_1$, and BbCheA$_2$ AlphaFold models reveals the extent by which BbCheA$_2$ P3 domain

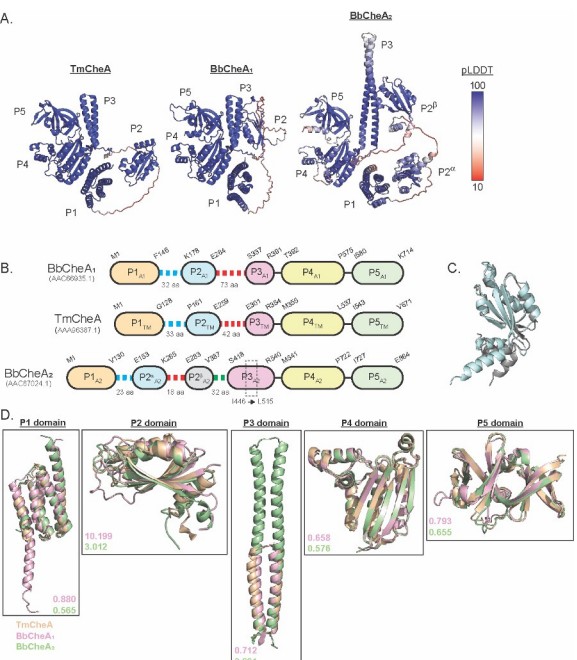

**Fig 1. Structural comparison of *B. burgdorferi* CheA₁ and CheA₂.** (A) AlphaFold models of *B. burgdorferi* CheA₁ (BbCheA₁) and CheA₂ (BbCheA₂) subunits compared to *Thermotoga maritima* CheA (TmCheA). Protein structures are colored according to their pLDDT scores [20]. (B) Domain organization of BbCheA₁, BbCheA₂, and TmCheA. The first P2 domain of BbCheA2 (referred to here as P2$^{\alpha}$) extends from Glu153 to Lys265 and is connected to the second P2 domain (P2$^{\beta}$) via an 18-residue long P2$^{\alpha}$-P2$^{\beta}$ linker. P2$^{\beta}$ is comprised of residues Glu283 to Val387 and is connected to the dimerization domain (P3) via a 32-residue long P2$^{\beta}$-P3 linker. Overall, the P1-P2$^{\alpha}$ and P2$^{\beta}$-P3 linkers consist of approximately 50 residues, similar to the 43-residue long P2-P3 linker of TmCheA. (C) Superimposition of BbCheA P2$^{\alpha}$ (RMSD: 0.48 Å, cyan, C150-V267) and P2$^{\beta}$ (gray, F279-V387). (D) Superimposition of phospho-transfer domain (P1), CheY-binding domain (P2), dimerization domain (P3), kinase domain (P4), and regulatory domain (P5) of TmCheA (tan), BbCheA₁ (pink), and BbCheA₂ (green). RMSD of each alignment is indicated in the lower region of each panel. All structural superimposition and figures were generated in PyMol [105].

differs from that of TmCheA and BbCheA₁, whose P3 domains align with an RMSD of 0.7 Å. Compared to TmCheA P3, BbCheA₂ P3 aligns with an RMSD of 3.7 Å and is approximately 43 Å longer. The extended P3 domain may be important for maintaining the proper protein-protein interactions between CheA subunits within a homodimer and/or between CheA and the chemosensory receptors in the arrays [21,24].

To better contextualize the differences in P2 and P3 predicted for the BbCheAs, we performed multiple-sequence alignment analysis on discrete groups of CheA sequences within the spirochete phylum. We first examined all the CheA sequences in spirochete species that contained only one CheA gene, assuming that this single CheA would be associated with chemotaxis signaling. Within this category, we examined only those members from the order *Treponematales*. Multiple sequence alignments of these CheA sequences reveal they have a single P2 domain and an extended P3 domain (**S2 Fig**). These results indicate that while the extra P2 domain may or may not play a role in chemotaxis for other spirochete species, the extended P3 domain is a conserved structural motif for CheAs associated with chemotaxis in spirochetes. We next looked at all CheA₁ and CheA₂ sequences from species in the *Borreliaceae* family. Multiple alignment data from these sequences reveal that in species with two distinct CheA isoforms, all the CheA₁ proteins encode a single P2 domain and have a short P3 domain similar to BbCheA₁ and TmCheA (**S3 Fig**), and all CheA₂ proteins encode for a second P2 domain and an extended P3 domain (**S3 Fig**). These data suggest that while an extended P3 domain

appears to be important for CheA proteins associated with chemotaxis in diverse spirochete species, the addition of a second P2 domain may be a *Borrelia* spp. and *Borreliella* spp. specific modification.

## Isolation of $cheA_1$ mutant and its complemented strain

To investigate the role of CheA$_1$ in the infectious cycle of *B. burgdorferi*, we have attempted to inactivate $cheA_1$ in the virulent B31 A3-68 *Δbbe02* strain [25]. pGA$_1$kan, a previously constructed vector [11], was linearized and electro-transformed into wild-type competent cells. A previously described PCR analysis was first carried out to screen for clones with the desired targeted mutagenesis [11]. One clone (referred to as $cheA_1^{mut}$) was selected for further analysis. For the complementation, the plasmid CheA$_1$/pBBE22 (**Fig 2A**) was electro-transformed into the $cheA_1^{mut}$ competent cells. The presence of CheA$_1$/pBBE22 in antibiotic-resistant colonies was confirmed by immunoblots using anti-CheA$_1$ antibody and anti-DnaK antibody as a control. As shown in Fig 2B, a band of approximately 80 kDa was detected in the wild type and the $cheA_1^{com}$ strain but was absent in $cheA_1^{mut}$, indicating that the cognate gene product was abrogated in the mutant and restored in the complemented strain (**Fig 2B**). *B. burgdorferi* B31 strain contains 21 linear and circular plasmids [26] of which some are essential for its infectivity but are easily lost during *in vitro* cultivation [27]. To check for the plasmid profile of our isolated strains, a previously developed PCR method was used to determine if the obtained positive clones contain the same plasmids as their parental strains [28]. Our results showed that $cheA_1^{mut}$ (**S4B Fig**) and its isogenic complemented strain (**S4C Fig**) contained the same plasmid profile as the wild-type strain (**S4A Fig**). Growth analysis indicated that deletion of $cheA_1$ did not affect the fitness of the mutant at 23˚C,34˚C or 37˚C (**Fig 2C–2E**) indicating that depletion of CheA$_1$ has no impact on *B. burgdorferi* growth *in vitro*.

## The cellular localization of CheA$_1$ is temporally and spatially regulated

As mentioned earlier, the genome of *B. burgdorferi* encodes for multiple copies of chemotaxis genes. It was speculated that *B. burgdorferi* may possess two chemotaxis pathways that function in different hosts during the infection cycle [8,11,13,18,29], i.e., CheA$_2$, CheW$_3$, and CheY$_3$ form a pathway that executes chemotaxis in mammalian hosts, whereas CheA$_1$, CheW$_2$, and CheY$_2$ form a distinct pathway that is activated in the tick vector. Consistent with this speculation, our previous work showed that inactivation of $cheA_2$ impaired the ability of the spirochete to establish infection in mice but not in ticks [17]. In order to decipher the function of CheA$_1$ in the enzootic cycle of *B. burgdorferi*, a GFP reporter construct (**S5A Fig**) was created and used to complement $cheA_1^{mut}$. The reporter strain was cultivated under laboratory conditions as well as conditions mimicking the tick milieu *in vitro* and observed microscopically. Surprisingly, we noticed that CheA$_1$-GFP was localized to one cell pole when cultivated under the tick-phase condition but appeared diffused at elevated temperature (**Fig 3A**). The diffused pattern was not due to degradation of the GFP-fusion protein at elevated temperature as full-length CheA$_1$-GFP fusion protein was detected with no excessive degradation (**S5C Fig**). This pattern was not observed in a strain carrying only the empty GFP vector (**Fig 3A**) or in a strain expressing CheA$_2$-GFP fusion protein (**S5C-S5D Fig**). This result indicates that the polar localization pattern is unique to CheA$_1$ (**Fig 3A**). In addition, the localization of CheA$_1$ appeared to be connected with cell division. During early log phase, the CheA$_1$-GFP signal was localized to one cell pole. As the cells entered mid log phase, the signal of CheA$_1$-GFP was seen distributed to the mid cell as well as both cell poles (**Fig 3B** Day 7 onward). As the cells entered late log to stationary phase, CheA$_1$-GFP signal was seen distributed along the cell body in specific puncta as indicated by the white arrows in Fig 3B Day 13–17 (**Fig 3B**). When the cells were cultivated

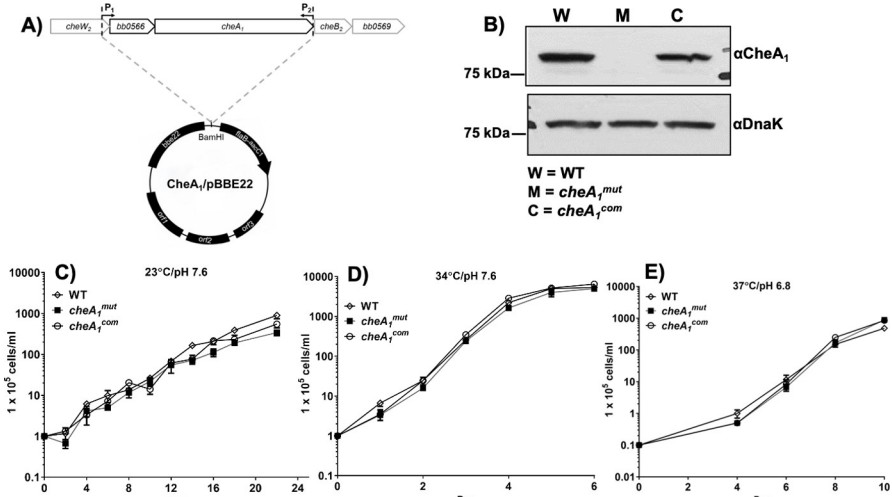

**Fig 2. Characterizations of *cheA₁* mutant (*cheA1mut*) and its isogenic complemented strain (*cheA1com*). (A)** Deletion of *cheA₁* has no impact on *B. burgdorferi* growth. Construction of CheA₁/pBBE22 for the complementation of *cheA₁ᵐᵘᵗ*. *bb0566-cheA₁* fragment was PCR amplified using primer pair P₁/P₂ and cloned into pBBE22 [17,99], a shuttle vector of *B. burgdorferi*, yielding CheA₁/pBBE22. **(B)** Immunoblot analysis of WT, *cheA₁ᵐᵘᵗ* and *cheA₁ᶜᵒᵐ* strains. The same amounts of WT, *cheA₁ᵐᵘᵗ* and *cheA₁ᶜᵒᵐ* whole-cell lysates were analyzed by SDS-PAGE and then probed with CheA₁ and DnaK antibodies, as previously documented [11,97]. Growth analysis of WT, *cheA₁ᵐᵘᵗ* and *cheA₁ᶜᵒᵐ* under routine laboratory condition **(C)** (34°C/pH7.6), **(D)** tick-like condition (23°C/pH7.6), and **(E)** host condition (37°C/pH6.8). Growth curve analysis was carried out to determine if CheA₁ affects cell growth. 10⁴ or 10⁵ cells/ml of bacteria were inoculated into 10 ml BSK-II media and cultivated under 34°C/pH7.6, 23°C/pH7.6, or 37°C/pH6.8. Cell numbers were enumerated every 1–4 days until cells entered stationary phase. Cell counting was repeated in triplicate with at least two independent samples, and the results are expressed as means ± SEM.

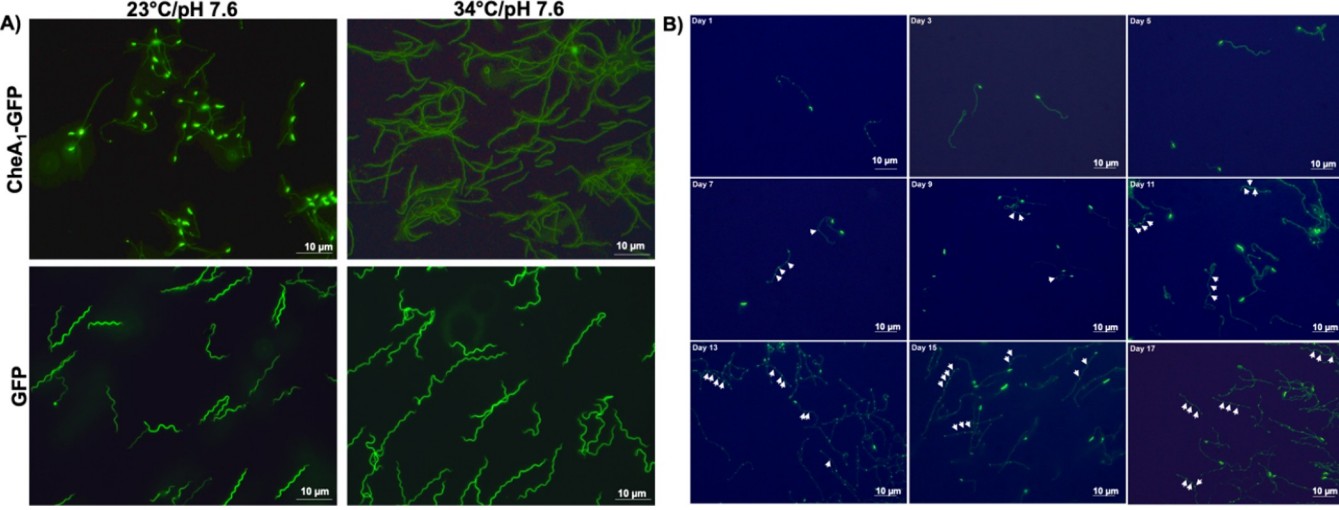

**Fig 3. CheA₁ has a unique polar localization when cultivated under tick-like condition. (A)** *In vitro* localization of CheA₁. To localize CheA₁, a CheA₁-GFP reporter construct (S3 Fig) was used to complement *cheA₁ᵐᵘᵗ*. The obtained cells were cultivated under unfed tick (UF) condition at 23°C/pH 7.6 or routine laboratory culture condition at 34°C/pH 7.6. Cells were monitored for the presence of GFP signal and images were taken at early log phase at ×200 magnification using a Zeiss Axiostar Plus microscope. Scale bars represent 10 μm. A *cheA₁ᵐᵘᵗ* carrying a GFP reporter construct was used as a control to confirm that the polar localization seen in CheA₁-GFP complemented strain is due to CheA₁ and not an artifact from GFP protein. **(B)** Time course images of CheA₁-GFP strain cultivated under UF condition. 10⁵ cells/ml of CheA₁-GFP strain was inoculated into 10 ml fresh BSK-II and cultivated under UF condition. Images were taken every 2 days to monitor the localization of CheA₁-GFP signals. As cells entered late log to early stationary phase, increasing numbers of discrete bright puncta were observed evenly distributed along the cell body, coincided with the zones of new peptidoglycan formation that mark the division sites for daughter cells [106].

under elevated temperature, the polar localization of CheA₁-GFP disappeared (**S6 Fig**). At elevated temperature, during the early growth phase, CheA₁-GFP signal appeared dispersed along the cell body and only when the cells entered stationary phase single pole localization reappeared as was the case with cells cultivated at 23˚C (**S6B Fig**), suggesting that the localization of CheA₁ is regulated spatially and temporally. In contrast to CheA₁-GFP, the localization of GFP and CheA₂-GFP remained evenly distributed along the cells under all culture conditions and growth phases (**S6A and S6C Fig**). Collectively, these results strongly suggest that the localization of CheA₁ is dynamic throughout the growth of *B. burgdorferi* and that CheA₁ function may be more prominent during the tick phase of the enzootic cycle. In addition, pH (acidic when cell density increases and enters stationary phase [30]) or temperature may have an impact on the localization and function of CheA₁.

## CheA₁ has no impact on *B. burgdorferi* swimming behavior and chemotaxis *in vitro*

*B. burgdorferi* has three different swimming modes: run, flex, and reverse [11,12]. Our previous work showed that deletion of *cheA₁* did not have impacts on the swimming behavior of *B. burgdorferi in vitro* [11]. All of these studies, however, were conducted using a high-passage avirulent B31A strain. To test whether this is the case in the virulent strain background, the swimming behaviors of the wild-type, *cheA₁^mut^*, and *cheA₁^com^* cells were analyzed using a computer-assisted cell tracker as documented previously [11]. Similar to the WT, *cheA₁^mut^* showed three swimming modes–runs interrupted by reverses/flexes, with a mean swimming velocity of 12.5 ± 0.5 μm/sec compared to 13.7 ± 0.3 and 12.9 ± 0.3 μm/sec of WT and *cheA₁^com^*, respectively (**Fig 4A**). In addition, capillary tube assays using *N*-acetyl-D-glucosamine (NAG) as a chemoattractant [31] (**Fig 4B**) and swimming plate analysis (**Fig 4C–4D**) did not reveal significant differences among WT, *cheA₁^mut^*, and *cheA₁^com^*, further supporting that CheA₁ is dispensable for *B. burgdorferi* motility and chemotaxis *in vitro*.

## *cheA₁^mut^* fails to establish systemic infection in immunocompetent mice by needle inoculation

To determine if CheA₁ is essential for the virulence of *B. burgdorferi* BALB/c mice were infected intradermally with 1 x 10⁵ cells of the WT, *cheA₁^mut^*, and *cheA₁^com^* strains (3–7 mice per strain). Tissues from the ear, skin, heart, and joint were harvested three weeks post-infection and transferred to BSK-II medium for re-isolation. The wild-type strain was successfully recovered from all of the infected animals (7/7) while *cheA₁^com^* was successfully recovered from some tissues of five infected mice (5/5), indicating a partial complementation. Interestingly, *cheA₁^mut^* was only recovered from the skin at the site of inoculation and not from distal organs (**Table 1**). To rule out the potential of a spontaneous mutation in our mutant, a new deletion construct was produced to replace the entire *cheA₁* gene in-frame with a kanamycin (*kan*) cassette (**S4D Fig**) and the mouse infection study was repeated. The newly constructed mutant (*cheA₁^IFD^*) was able to survive at the inoculation site but failed to establish systemic infection in immunocompetent mice (**Table 1**), similar to *cheA₁^mut^*. Taken together, these results suggest that CheA₁ is not essential for *B. burgdorferi* colonization of skin, but is required for dissemination from the initial inoculation site and/or to establish a systemic infection in mice.

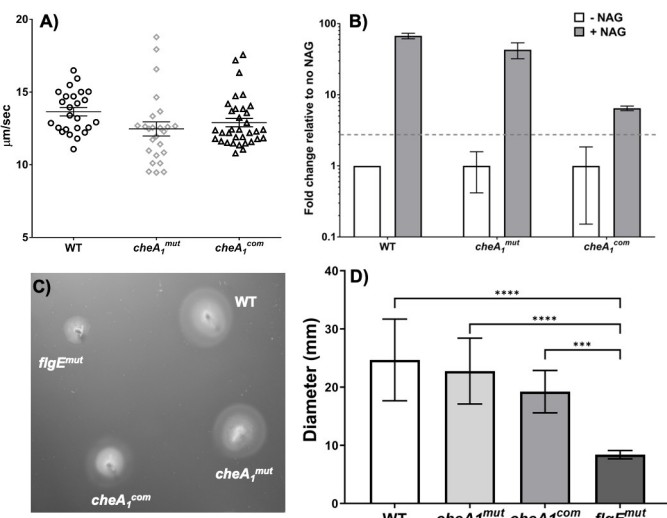

**Fig 4. CheA$_1$ is dispensable for spirochete motility and chemotaxis *in vitro*. (A)** Bacteria tracking analysis. The swimming velocities of WT, *cheA$_1$*$^{mut}$, and *cheA$_1$*$^{com}$ were measured using a computer-based motion tracking system as previously described [11]. At least 25 cells were tracked and the swimming velocities were measured. **(B)** Capillary tube chemotaxis assay. This experiment was carried out using 100 mM *N*-acetyl-D-glucosamine (NAG) as an attractant according to a previous report [31]. The results are expressed as fold increases in cell number entering the capillary tubes containing the attractant relative to the number entering control tubes without attractant (buffer alone). Results are expressed as means ± SEM for five tubes. A 2-fold increase is considered the threshold for a positive response. **(C)** and **(D)** Swimming plate assays. The assay was carried out on 0.35% agarose containing 1:10-diluted BSK-II medium as described previously [100]. The *flgE* mutant, a previously constructed nonmotile mutant [103], was included to determine the sizes of the inocula. The data are presented as mean diameters (in millimeters) of rings ± SEM for twelve plates. All assays were repeated at least twice with independent replicates and representative data are shown here. * significant difference (*P* < 0.05).

## *cheA$_1$*$^{mut}$ can persist and proliferate at the site of inoculation

Intradermal infection of the mutant failed to cause systemic infection but the mutant could be re-isolated from the initial inoculation site, suggesting that the mutant is able to survive and persist at the site of injection and evade the host immune system. To determine how long the mutant can persist at the inoculation site, 1 x 10$^5$ cells of WT or *cheA$_1$*$^{mut}$ were injected subcutaneously into BALB/c mice and the inoculation site was marked for bacterial re-isolation at different time points as indicated in the Material and Methods. qRT-PCR analysis showed that the mutant was not only able to persist but replicate at the inoculation site to the same extent as the WT as indicated by the increase in *flaB* transcript copies for up to 28 days (**Fig 5**).

**Table 1. *cheA1mut* is unable to cause systemic infection in immunocompetent mice.**[a]

| Mouse strain | *Bb* Strains | No. of cultures positive/Total no. specimens examined | | | | |
|---|---|---|---|---|---|---|
| BALB/c | | Skin (inoculation site) | Ear | Heart | Joint | No. of mice infected/ total no. of mice used |
| | WT | 4/7 | 7/7 | 7/7 | 7/7 | 7/7 |
| | *cheA$_1$*$^{mut}$ | 4/7 | 0/7 | 0/7 | 0/7 | 4/7 |
| | *cheA$_1$*$^{IFD}$ | 3/3 | 0/3 | 0/3 | 0/3 | 3/3 |
| | *cheA$_1$*$^{com}$ | 1/5 | 3/5 | 2/5 | 1/5 | 5/5 |

[a] Groups of three to seven BALB/c mice were intradermally inoculated with 10$^5$ spirochetes of WT, *cheA$_1$*$^{mut}$, *cheA$_1$*$^{IFD}$, or *cheA$_1$*$^{com}$ strains. *cheA$_1$*$^{IFD}$ is an in-frame deletion mutant of *cheA$_1$* where the entire open reading frame of *cheA$_1$* is replaced with a *kan* cassette (**S4D Fig**). Mice were sacrificed 3 weeks post-inoculation; skin (inoculation site), ear, heart, and joint specimens were harvested for spirochete recovery in BSK-II medium.

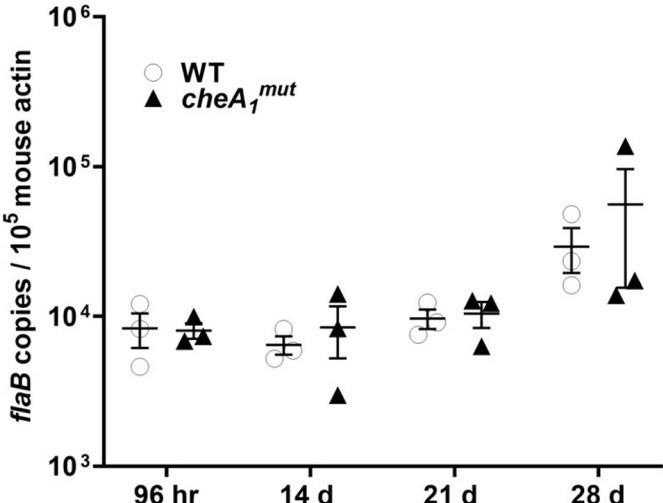

**Fig 5. *cheA1^mut* is able to persist at the site of inoculation.** To determine if CheA$_1$ is required for *B. burgdorferi* persistence at the inoculation site, a time point infection study was performed. 18 immunocompetent mice were subcutaneously infected with $10^5$ of WT or *cheA$_1$^mut* and mice were sacrificed at the indicated time points. Skin tissue from the inoculation sites were recovered to assess for borrelial burden using qRT-PCR as described [17]. The data are presented as mean *flaB* transcript copies over $10^5$ of mouse *β-actin* ± SEM for each group of mice.

## *cheA$_1$^mut* is defective in hematogenous dissemination

We reasoned that the failure of the mutant to cause systemic infection could be due to the following three possibilities: 1) the mutant failed to disseminate from the injection site; 2) the mutant successfully disseminated from the injection site into the bloodstream but failed to spread to distal organs due to a defect in tissue tropism; or 3) the mutant successfully disseminated into the bloodstream but was cleared by the host immune system. To test these possibilities, the above animal study was repeated using severe combined immuno-deficiency (SCIDs) mice. Based on the tissue re-isolation results in Table 2, WT and *cheA$_1$^com* were successfully recovered from all of the harvested tissues. In contrast, the *cheA$_1$^mut* mutant was able to disseminate and cause systemic infection in 50% of the injected SCID mice. Consistent with the result obtained using immunocompetent mice, *cheA$_1$^mut* was able to colonize and persist in the skin better than in other tissues (**Table 2**). qRT-PCR analysis indicated that of those 50% SCID mice with systemic infection, no statistically significant difference in tissue spirochetal burdens was detected between the wild type, *cheA$_1$^mut*, and *cheA$_1$^com* with the exception of the ear and heart tissues (**Fig 6**). This result strongly suggests that in the absence of an adaptive immune response, *cheA$_1$^mut* can disseminate from the injection site and cause systemic infection albeit

**Table 2. *cheA$_1$^mut* is still attenuated in distal organ colonization in SCID mice.[b]**

| Mouse strain | *Bb* Strains | No. of cultures positive/Total no. specimens examined | | | | | |
|---|---|---|---|---|---|---|---|
| SCID | | Skin | | Ear | Heart | Joint | Blood |
| | | (inoculation site) | (distal) | | | | |
| | WT | 6/6 | 6/6 | 6/6 | 6/6 | 6/6 | 6/6 |
| | *cheA$_1$^mut* | 6/6 | 5/6 | 3/6 | 3/6 | 3/6 | 3/6 |
| | *cheA$_1$^com* | 3/3 | 3/3 | 3/3 | 3/3 | 3/3 | 3/3 |

[b] Groups of three to six SCID mice were intradermally inoculated with $10^5$ spirochetes of WT, *cheA$_1$^mut*, and *cheA$_1$^com* strains. Mice were sacrificed 3 weeks post-inoculation; skin (inoculation and distal site), ear, heart, joint, and blood specimens were harvested for spirochete culture in BSK-II medium.

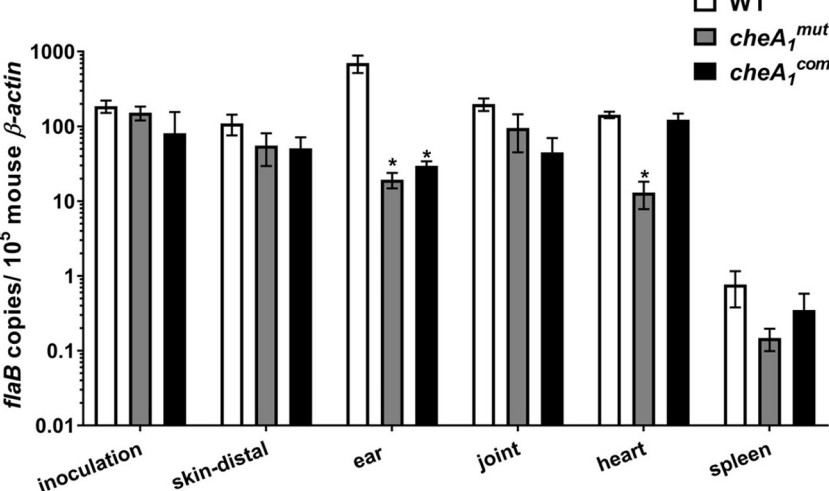

**Fig 6. *cheA₁ᵐᵘᵗ* is attenuated in causing systemic infection in immunodeficient mice.** To determine if CheA₁ is required for *B. burgdorferi* immune evasion, needle infection study was repeated using severe combined immunodeficiency (SCIDs) mice. For this study, $10^5$ of WT, *cheA₁ᵐᵘᵗ* and *cheA₁ᶜᵒᵐ* strains were subcutaneously inoculated into SCID mice and sacrificed three weeks after infection. Skin tissues around and distal from the injection site were harvested along with tissues from the ear, joint, heart and spleen to assess for borrelial burden using qRT-PCR as described [17]. The data are presented as mean *flaB* transcript copies over $10^5$ of mouse *β-actin* ± SEM. *, significant difference ($P < 0.05$).

to an attenuated degree. In the immunocompetent mice, *cheA₁ᵐᵘᵗ* most likely failed to evade the host adaptive immune response and was cleared from the animal before it was able to colonize and establish infection in distal organs. Furthermore, blood culture showed that 3 out of 6 mice were positive for *cheA₁ᵐᵘᵗ* when injected subcutaneously into SCID mice (**Table 2**), further substantiating that the mutant is able to disseminate from the inoculation site and spread systemically through the bloodstream. Taken together, the mouse infection studies indicate that in the absence of CheA₁, *B. burgdorferi* is attenuated in both innate and adaptive immune evasion and tissue colonization.

## *cheA₁ᵐᵘᵗ* is attenuated in colonization and adaptive immune evasion

The above study showed that the mutant has reduced infectivity but is still able to disseminate into the bloodstream to cause systemic infection in SCID mice but not in immunocompetent mice. It is possible that in the presence of adaptive immunity, the low number of mutant cells that survived was unable to disseminate and cause systemic infection. To rule out this possibility, a 10-fold higher number ($1 \times 10^6$ cells) of spirochetes were directly introduced into the

**Table 3. *cheA₁ᵐᵘᵗ* has defects in dissemination and immune evasion.ᶜ**

| Mouse strain | *Bb* Strains | No. of cultures positive/Total no. specimens examined | | | | |
|---|---|---|---|---|---|---|
| | | Skin | Ear | Heart | Joint | Blood |
| BALB/c | | | | | | |
| | WT | 2/3 | 2/3 | 2/3 | 3/3 | 3/3 |
| | *cheA₁ᵐᵘᵗ* | 1/5 | 0/5 | 0/5 | 0/5 | 0/5 |
| | *flaBᵐᵘᵗ* | 0/3 | 0/3 | 0/3 | 0/3 | 0/3 |

ᶜ Groups of three to five BALB/c mice were tail-vein injected with $10^6$ spirochetes of WT, *cheA₁ᵐᵘᵗ* or *flaBᵐᵘᵗ* strains. Mice were sacrificed 2 weeks post-inoculation; skin, ear, heart, joint, and blood specimens were harvested for spirochete recovery in BSK-II medium.

bloodstream of BALB/c mice via tail-vein injection to determine if high doses of bacteria can overcome the dissemination and colonization barrier. As a control, a non-motile, non-infectious $flaB^{mut}$ strain was included [32]. 1 x $10^6$ of WT, $cheA_1^{mut}$ or $flaB^{mut}$ cells in 100 μl of PBS were injected into the tail-vein of BALB/c mice. Tissues were harvested two weeks post-injection to check for dissemination. All mice injected with the WT strain showed disseminated infection (3/3) whereas the $flaB^{mut}$ control was non-infectious as expected (0/3). For mice injected with the $cheA_1^{mut}$ strain, live spirochetes were successfully recovered from the skin of only 1 out of 5 mice infected, indicating that the mutant is attenuated in tissue colonization even at a high inoculum introduced directly into the bloodstream (**Table 3**). In addition, no mutant cells were recovered from the blood of any mice injected with $cheA_1^{mut}$. This result clearly indicates that CheA$_1$ is essential for the survival of *B. burgdorferi* in the blood. Consistent with live culture recovery, high levels of *flaB* transcripts were detected from mice infected with the WT strain whereas $cheA_1^{mut}$-infected mice had less than one copy of *flaB* transcript per $10^5$ of mouse *β*-actin transcript overall (most likely due to residual *B. burgdorferi* that was initially introduced into the bloodstream (**Fig 7A**)). These findings further substantiate the key role of CheA$_1$ in *B. burgdorferi* immune evasion and tissue colonization.

## $cheA_1^{mut}$ is more susceptible to serum bactericidal activity

*B. burgdorferi* is able to resist the bactericidal activity of serum [33,34] as it produces several virulence factors that protects the spirochetes from complement mediated killing, such as BBK32, OspC, and CRASP proteins [see recent reviews [35–38]]. The attenuated infectivity of $cheA_1^{mut}$ in both the SCID mice and tail-vein infection model suggests that CheA$_1$ is important for the immune evasion of *B. burgdorferi*. To determine if deletion of CheA$_1$ renders the spirochete susceptible to serum bactericidal activity, WT, $cheA_1^{mut}$ and $cheA_1^{com}$ were treated with normal human serum (NHS) or heat-inactivated NHS (hiNHS) and the percentage of viable cells after treatment was normalized against the input cells. Our data showed that $cheA_1^{mut}$ was more susceptible to serum killing with ~25% survival rate as compared to ~60% survival rate of the WT and $cheA_1^{com}$ in NHS (**Fig 7B**). This observation is in congruence with the SCID mice study where only 50% of $cheA_1^{mut}$ was able to survive in the blood of infected mice and the complemented strain fully restored the survival rate to WT level (**Table 2**). The reduced survival of the mutant in the bloodstream can likely be attributed to its susceptibility to serum bactericidal activity.

## CheA$_1$ is not required for colonization of *Ixodes scapularis* ticks

As $cheA_1^{mut}$ failed to cause disseminated infection in wild-type mice, we used a microinjection-based infection procedure to artificially infect ticks [39]. For this study, Naïve *I. scapularis* nymphs were microinjected with equal amounts of WT, $cheA_1^{mut}$, or $cheA_1^{com}$ strains. After the infection, ticks were allowed to feed to repletion on naïve C3H mice (5 ticks/ mouse and 3 mice for each strain). The spirochete burdens in engorged ticks were measured by qRT-PCR as described previously [17,40,41]. As shown in Fig 8, the bacterial burden of $cheA_1^{mut}$ in fed ticks was not significantly different from that of the WT and $cheA_1^{com}$ (**Fig 8A**). This result indicates that CheA$_1$ is not required for *B. burgdorferi* colonization and survival in the tick vector.

## $cheA_1^{mut}$ fails to be transmitted to mice via tick bite

The infectivity of $cheA_1^{mut}$ in mice was first evaluated via needle inoculation. However, this method is different from the natural route of mammalian infection through tick bite. As $cheA_1^{mut}$ is still able to establish infection in ticks (**Fig 8A**), we examined whether the mutant can be transmitted to mice when fed upon by infected ticks. Groups of naïve C3H mice were

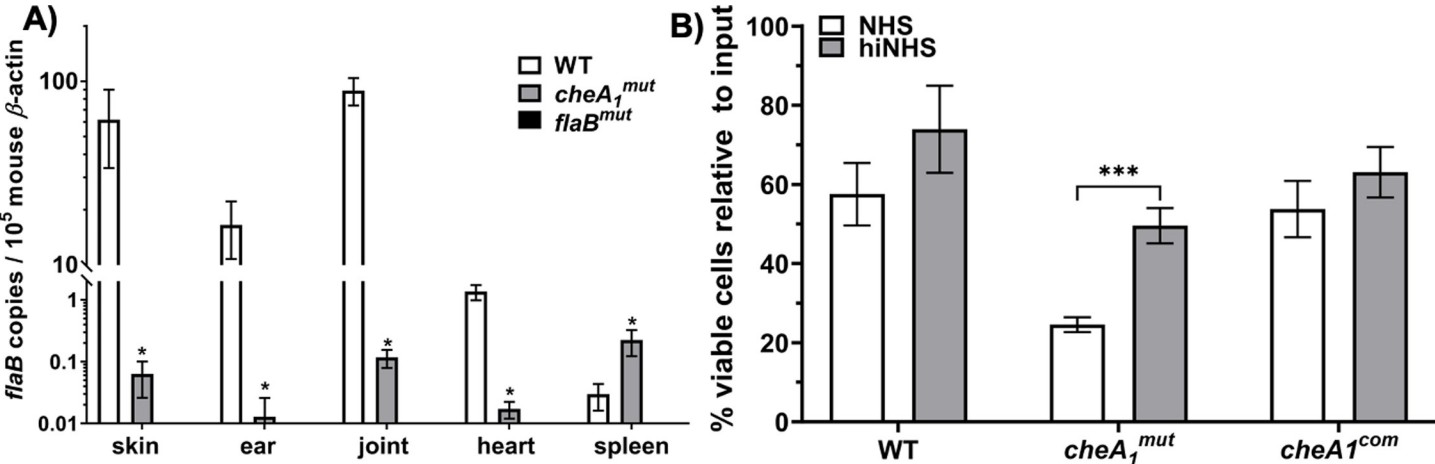

**Fig 7. *cheA₁ᵐᵘᵗ* has defects in dissemination and serum resistance. (A)** To determine if high dose of inoculum can restore *cheA₁ᵐᵘᵗ* immune evasion and colonization barriers, $10^6$ of WT, *cheA₁ᵐᵘᵗ* and non-motile *flaBᵐᵘᵗ* [32] strains were injected into wild type mice via tail-vein and sacrificed two weeks after infection. Tissues from the skin, ear, joint, heart and spleen were harvested to assess for borrelial burden using qRT-PCR as described [17]. The data are presented as mean *flaB* transcript copies over $10^5$ of mouse *β-actin* ± SEM. **(B)** Serum bactericidal activity. $10^6$ cells of WT, *cheA₁ᵐᵘᵗ* and *cheA₁ᶜᵒᵐ* were incubated with 20% normal human serum (NHS) or heat-inactivated serum (hiNHS) for 2 hours. The percentage of viable cells in NHS treated cells were determined after normalization to their respective hiNHS-treated groups. The data shown here are means of three independent replicates ± SEM. *, significant difference ($P < 0.05$).

allowed to be parasitized by artificially infected ticks as detailed in the Material and Methods. Seven days after tick feeding, the mice were sacrificed and specimens from the skin, ear, heart, bladder and blood were collected and subjected to qRT-PCR analysis to determine the pathogen level. Consistent with the results obtained via needle inoculation (**Table 1**), only the tissue specimens from mice fed on by WT- and *cheA₁ᶜᵒᵐ*-infected ticks were positive for *flaB* transcripts via qRT-PCR analysis (**Table 4**). *flaB* transcripts were only detected from the skin and heart of one mouse fed upon by *cheA₁ᶜᵒᵐ*-infected ticks, consistent with the results observed by needle inoculation of mice in which only a partial complementation was achieved with

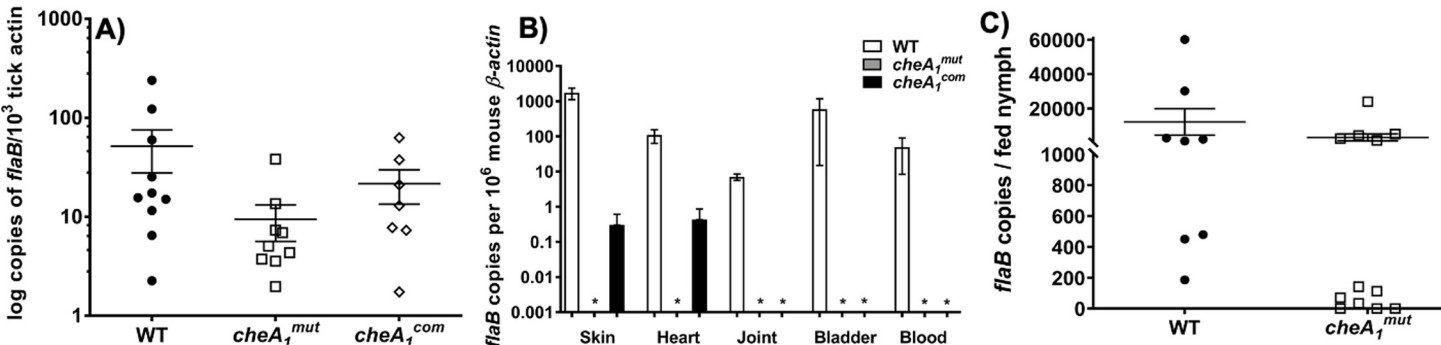

**Fig 8. CheA₁ is not required for *B. burgdorferi* acquisition and survival in tick but necessary for transmission. (A)** Detection of spirochete burdens in microinjected nymphal ticks after feeding. RNA samples were extracted from whole fed ticks (after repletion; 5 to 7 days) and subjected to qRT-PCR analysis. The bacterial burdens in ticks were measured by the number of copies of *flaB* transcript compared to the number of copies of tick *β-actin* transcript as previously described [17]. The data are presented as the means of relative levels of *flaB* transcript ± SEM for each strain (WT, *cheA₁ᵐᵘᵗ*, and *cheA₁ᶜᵒᵐ*). **(B)** Detection of spirochete burdens in mice infected via tick bite. At day 14 after tick feeding, mice were sacrificed and tissues from the skin, heart, joint, and bladder were harvested for qRT-PCR analysis as previously documented [17,107]. No trace of *flaB* transcript was detected in the mouse tissues fed upon by *cheA₁ᵐᵘᵗ* infected ticks while 1 in 3 mice fed upon by *cheA₁ᶜᵒᵐ* showed positive results at the skin and heart tissues. *, significant difference ($P < 0.05$). **(C)** Detection of spirochete burdens in naïve nymphal ticks fed on infected mice. C3H mice were artificially infected with WT or *cheA₁ᵐᵘᵗ* strain via needle inoculation. Naïve nymphal ticks were confined to the injection site and allowed to feed to repletion. After 72 hours, fed nymphs were collected and individually tested for the presence of *flaB* via qPCR. Experiments were repeated twice and data are presented as mean *flaB* copies per fed nymph from both data sets ± SEM.

**Table 4. $cheA_1^{mut}$ has defects in tick transmission.[d]**

| *Bb* Strains | No. of *flaB* positive tissues/Total no. specimens examined | | | | |
|---|---|---|---|---|---|
| | Skin | Heart | Joint | Bladder | Blood |
| WT | 3/3 | 3/3 | 3/3 | 3/3 | 3/3 |
| $cheA_1^{mut}$ | 0/3 | 0/3 | 0/3 | 0/3 | 0/3 |
| $cheA_1^{com}$ | 1/3 | 1/3 | 0/3 | 0/3 | 0/3 |

[d] Naïve *Ixodes scapularis* nymph were microinjected with ~ 5 x 10⁶ spirochetes of WT, $cheA_1^{mut}$ or $cheA_1^{com}$ strains. Two days after injection, infected nymphs were allowed to parasitize on naïve C3H mice (3 mice each group) until repletion. Mice were sacrificed 2 weeks post-feeding; skin, heart, joint, bladder, and blood specimens were harvested for *flaB* detection via qRT-PCR.

$cheA_1^{com}$ (**Table 1**). No *flaB* transcript was detected in the specimens from mice fed upon by $cheA_1^{mut}$-infected ticks (**Table 4**), indicating that the mutant failed to be transmitted to mice even though it could colonize the tick vector. Collectively, these results suggest that CheA₁ is essential for the transmission of *B. burgdorferi* to mice via tick bite.

## $cheA_1^{mut}$ can be acquired by naïve tick from infected mice

The tick transmission experiment revealed that $cheA_1^{mut}$ was able to survive in ticks via artificial infection but failed to transmit to naïve mammalian hosts (**Fig 8A and Table 4**). To test the possible role of CheA₁ in the acquisition phase, mice were infected with WT or $cheA_1^{mut}$ via needle inoculation. Two weeks post-injection, infection was confirmed via culturing of the skin biopsy from the injection site. Nymphs were placed and confined to the injection site and allowed to feed for 72 hours. Fed ticks were then collected for qPCR to determine the spirochete acquisition efficiency. As shown in **Fig 8B**, the *flaB* gene was successfully detected in nymphs that parasitized mutant-infected mice and its copy number was comparable to that detected in nymphs that fed on WT-infected mice (**Fig 8C**). It should be noted that in the majority of $cheA_1^{mut}$-infected ticks, acquisition was greatly attenuated relative to wild type-infected nymphs with 3 out of 12 nymphs showing no detectable *flaB* gene. This result suggests that CheA₁ is not absolutely required for spirochete acquisition by ticks from mice, but may play some role in the process.

## CheA₁ affects expression of the RpoS-regulon and RpoS stability

*B. burgdorferi* encodes several proteins that are essential for host tissue binding/interaction and immune evasion in mammalian hosts, e.g., OspC, BBK32, and DbpB/A [42–45]. The attenuated phenotype of $cheA_1^{mut}$ observed in the mouse infection studies could be the result of altered expression of virulence-associated proteins. To test this possibility, *B. burgdorferi* cells were cultivated under elevated temperature and reduced pH (37˚C/pH 6.8) to mimic the host environment. Total protein profiles of WT, $cheA_1^{mut}$ and $cheA_1^{com}$ were analyzed on SDS-PAGE (**Fig 9A**) and a number of *B. burgdorferi* virulence determinants were measured by immunoblotting analysis and compared to the level of DnaK, a loading control. The protein level of RpoS was significantly reduced in the $cheA_1^{mut}$ mutant as compared to the WT and $cheA_1^{com}$ strains (**Fig 9B**). Consistent with this finding, the levels of several RpoS-regulated proteins (i.e., BBK32, OspC, and DbpA) were also significantly reduced in the mutant and restored in the complemented strain. By contrast, the level of P66, a non-RpoS regulated protein [46], was not affected by the deletion of $cheA_1$, suggesting that CheA₁ specifically affects RpoS and RpoS-regulon expression (**Fig 9B**). qRT-PCR analysis revealed that the transcript level of *rpoS* was slightly reduced in the mutant but not significant while the transcript levels of

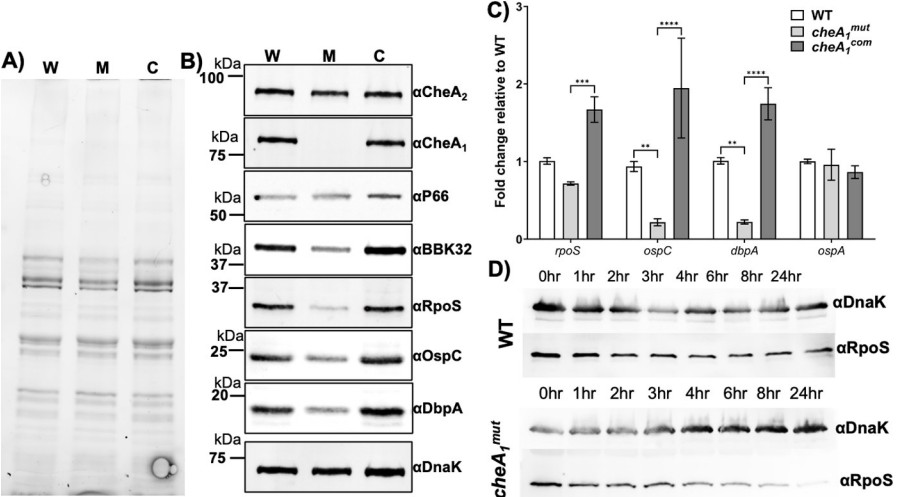

**Fig 9. Depletion of CheA₁ negatively affects the level of RpoS, BBK32, OspC, and DbpA.** *B. burgdorferi* strains (WT, *cheA₁ᵐᵘᵗ*, and *cheA₁ᶜᵒᵐ*) were cultivated at 37°C/pH 6.8 to mimic the mammalian host condition and harvested at stationary phase (∼10⁸ cells/ml) for **(A,B)** immunoblotting analysis, and **(C)** qRT-PCR analysis. **(A)** Similar amounts of whole-cell lysates were analyzed by stain-free SDS-PAGE. Gel image showed the total protein of WT, *cheA₁ᵐᵘᵗ*, and *cheA₁ᶜᵒᵐ* analyzed on SDS-PAGE. **(B)** Immunoblot analysis of WT, *cheA₁ᵐᵘᵗ*, and *cheA₁ᶜᵒᵐ* samples using specific antibodies against CheA₂, CheA₁ [11], P66 [101], BBK32 [102], RpoS, OspC, and DbpA [86]. DnaK was used as an internal control, as previously described [97]. **(C)** Detection of *rpoS*, *ospC*, *dbpA*, and *ospA* transcript level in WT, *cheA₁ᵐᵘᵗ*, and *cheA₁ᶜᵒᵐ* using qRT-PCR. The *dnaK* transcript was used as an internal control. Experiments were repeated at least twice with independent replicates. Data from two replicates are expressed as mean fold change relative to WT ± SEM. *, significant difference ($P < 0.05$). **(D)** RpoS protein turnover in WT and *cheA₁ᵐᵘᵗ*. The stability of RpoS protein was examined in stationary phase WT B31 A3-68 and B31 A3-68 *cheA₁ᵐᵘᵗ* upon protein synthesis arrest using spectinomycin. Samples were harvested at the indicated time points and probed using antibody against RpoS and DnaK (as a loading control). Experiments were repeated at least three times with independent biological replicates. A representative image is shown here.

*ospC* and *dbpA* were significantly reduced by 65% and 75%, respectively (**Fig 9C**). In contrast, no significant change was observed in the mutant for *ospA* transcript, a RpoD-regulated gene [47]. In sum, deletion of *cheA₁* leads to reduced expression of RpoS and RpoS-regulated virulence factors, which in turn could explain the attenuated infectivity observed with *cheA₁ᵐᵘᵗ*. Depletion of CheA₁ affects RpoS post-transcriptionally as no significant reduction in *rpoS* transcript was detected in *cheA₁ᵐᵘᵗ* (**Fig 9C**). To determine how CheA₁ affects RpoS protein expression, protein turnover assays were performed using WT and mutant cells cultivated at the condition mimicking the host environment to induce RpoS expression. Compared to the wild type, RpoS turnover was increased in the absence of CheA₁. The level of RpoS was relatively stable in the stationary phase of the wild-type culture whereas over 50% turnover of RpoS was observed in *cheA₁ᵐᵘᵗ* as early as one hour after protein synthesis was arrested (**Fig 9D**). Complementation of CheA₁ successfully restored the stability of RpoS (**S7 Fig**). Thus, the reduced RpoS protein level observed in *cheA₁ᵐᵘᵗ* is due to reduced protein stability in the absence of CheA₁.

## Depletion of CheA₁ leads to dysregulated expression of proteases

The homeostasis and stability of proteins are governed by a set of proteases responsible for the regulated degradation of a given protein target [48,49]. The genome of *B. burgdorferi* encodes several proteases, including Lon-1 [50], Lon-2 [51], HtrA [52–55], CtpA [56], and Clp [9]. Lon-1 protease was shown to negatively affect RpoS expression at the transcriptional level [50] while Lon-2 protease positively regulates RpoS expression at the post-transcriptional level

[51]. In other bacteria, such as *E. coli*, the protein stability of RpoS is governed by the ClpXP protease [57–59]. Since the stability of RpoS protein was affected in *cheA₁ᵐᵘᵗ*, the expression of these known proteases was examined in the mutant. qRT-PCR data from two independent biological replicates consistently showed that the transcripts of *clpX*, *lon-1*, and *lon-2* were significantly reduced in *cheA₁ᵐᵘᵗ* as compared to the WT, which was successfully restored in the complemented strain (**Fig 10A**). Preliminary RNA-seq analysis also revealed a significant down-regulation of *clpX*, *lon-1* and *lon-2* expression in *cheA₁ᵐᵘᵗ* (**S8 Fig**). Our data suggest that depletion of CheA₁ leads to dysregulated expression of *B. burgdorferi* proteases, such as *clpX* and *lon*, which subsequently affects the expression and protein stability of RpoS in a negative manner. In congruence with this, deletion of *clpX* significantly impaired the expression of RpoS along with the various RpoS-regulated proteins (**Fig 10B**) while deletion of *lon-1* led to a high level of RpoS expression as shown by Thompson *et al.* [50], further supporting the regulatory role of ClpX and Lon proteases on RpoS expression.

## *cheA₁ᵐᵘᵗ* fails to elicit mouse *il-10, tnf-α, ccl2,* and *il-1β* expression

The attenuated phenotype of *cheA₁ᵐᵘᵗ* in establishing systemic infection even in immunodeficient mice implicates a defect in both innate and adaptive immune evasion. Interleukin 10 (IL-10) is a key mediator in innate immune responses during Lyme disease infection [60]. Timely induction of IL-10 expression is critical for the disease progression by dampening the host inflammatory response to favor the survival of *B. burgdorferi* during early infection. To dissect the cause behind *cheA₁ᵐᵘᵗ* failure to establish systemic infection in mice, RNA-seq analysis was performed using infected mouse skin tissues. Briefly, 100 μl of 1 x 10⁷ WT or *cheA₁ᵐᵘᵗ* were needle inoculated into the dorsal side of BALB/c mice at three separate locations. Skin tissues were harvested from the inoculation sites at 72-hour post-injection (p.i.) for RNA extraction and RNA-seq analysis. Scatter plot of enriched KEGG pathway showed that distinct host genes were differentially induced between Sham vs WT and Sham vs MT (*cheA₁ᵐᵘᵗ*) infected samples at 72 hrs p.i. (**Fig 11A**). In WT-infected samples, significant changes of numerous

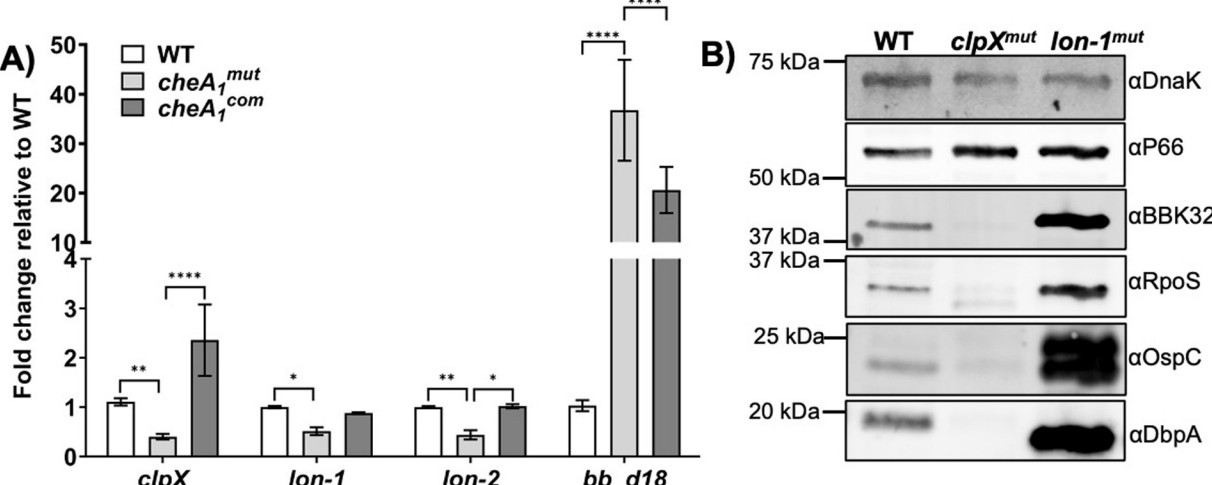

**Fig 10. Depletion of CheA₁ leads to dysregulated expression of proteases. (A)** Detection of *clpX*, *lon-1*, and *lon-2* transcript level in WT, *cheA₁ᵐᵘᵗ*, and *cheA₁ᶜᵒᵐ* using qRT-PCR. The *dnaK* transcript was used as an internal control. Experiment was repeated with two independent replicates. Data are expressed as mean fold change relative to WT ± SEM. *, significant difference ($P < 0.05$). **(B)** Immunoblot analysis of WT, *clpXᵐᵘᵗ*, and *lon-1ᵐᵘᵗ* samples using specific antibodies against P66 [101], BBK32 [102], RpoS, OspC, and DbpA [86]. DnaK was used as an internal control, as previously described [97]. Experiments were repeated twice with independent replicates. Lon-1 mutant (*lon-1ᵐᵘᵗ*) was kindly provided by the Ouyang's lab as a positive control to show the up-regulation of RpoS as previously described [50].

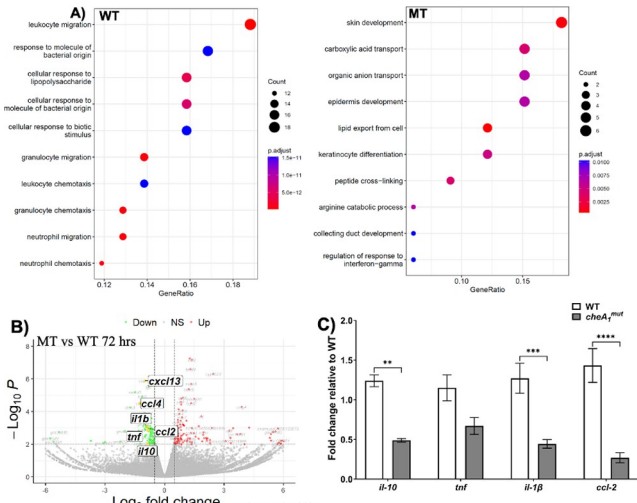

**Fig 11. RNA-seq analysis of BALB/c mouse skin tissues infected by WT and *cheA₁ᵐᵘᵗ*. (A)** Scatter plot of enriched KEGG pathway of differentially expressed genes between Sham and infected samples at 72hr p.i. Vertical coordinates represent pathway name, and horizontal coordinates represent Rich factor. The size and color of point represent the number of differential genes in the pathway and the range of different Q value, respectively. **(B)** Volcano plot of differentially expressed genes (DEG) between MT (*cheA₁ᵐᵘᵗ*) vs WT samples at 72hr p.i.. Each dot represents a gene, and its position is determined by the log2 fold change (x-axis) and the -log10 (*p*-value) (y-axis). Genes with significant upregulation (red) or downregulation (green) are highlighted, while non-significant genes are shown in gray. **(C)** Detection of *il-10*, *tnf*, *il-1β*, and *ccl-2* transcript levels in WT and *cheA₁ᵐᵘᵗ* infected mouse skin tissues using qRT-PCR. The *β-actin* transcript was used as an internal control. Data are expressed as mean fold change relative to WT ± SEM. *, significant difference ($P < 0.05$).

immune system related genes were observed, such as genes involved in leukocyte, granulocyte, and neutrophil migration. These genes were unchanged in the mutant infected samples. The transcript levels of several important cytokines and chemokines that have been reported to contribute to Lyme disease progression, such as IL-10, IL-1β, TNF-α, and CCL2 [61–68], were upregulated in the WT- but not MT-infected tissues (**Fig 11B**), which was validated by qRT-PCR (**Fig 11C**). Consistent with the RNA-seq analysis, qRT-PCR showed that the transcript levels of *il-10*, *il-1β*, and *ccl-2* were significantly lower in the *cheA₁ᵐᵘᵗ*-infected mouse skin tissues as compared to those infected by WT (**Fig 11C**). The data obtained from RNA-seq analysis suggests that the inability of *cheA₁ᵐᵘᵗ* to establish systemic infection is partly due to its failure in eliciting timely expression of key host cytokines and chemokines to aid in dissemination and colonization.

## Discussion

*B. burgdorferi* lives a complex enzootic life cycle alternating between arthropod vector and mammalian reservoir. As such, highly coordinated expressions of tick- and host-specific transcriptomes are indispensable to ensure the survival and persistence of the spirochete in nature. Two two-component systems (TCS), the Hk2-Rrp2-RpoN-RpoS and HK1-Rrp1 signaling pathways, are largely responsible for these extensive transcriptional adjustments (see reviews [69–71]). Among the enzootic cycle stage-specific differentially expressed genes, in addition to multiple crucial surface lipoproteins, many chemotaxis-related genes are also found to have distinct expression profiles when *B. burgdorferi* alternates between different hosts, which strongly indicates that the composition of the *B. burgdorferi* chemotaxis machinery differs between ticks and mammals [72]. The genome of *B. burgdorferi* encodes for orthologs of chemotaxis-related genes, many of which are present in several copies, such as two chemotaxis histidine kinases

($cheA_1$, $cheA_2$) and three chemotaxis response regulators ($cheY_1$, $cheY_2$, $cheY_3$) [9]. Some of these genes have no known function in chemotaxis as mutagenesis studies revealed no apparent contribution to chemotaxis *in vitro*. For example, a mutant deficient in $CheA_1$ exhibited a normal chemotaxis response whereas loss of $CheA_2$ resulted in a non-chemotactic phenotype *in vitro* [11,17]. Hence, understanding how these chemotaxis-related proteins are coordinated during the natural enzootic cycle of *B. burgdorferi* is crucial to fully comprehend the significance of each protein in the modulation of chemotaxis responses *in vivo*.

Interestingly, even though $CheA_1$ is not required for chemotaxis *in vitro*, it has a domain arrangement and a local domain architecture more similar to TmCheA than the chemotactic $CheA_2$. AlphaFold models reveal that $CheA_1$ has a single P2 domain, while $CheA_2$ has two P2 domains (α and β, **Fig 1A–1B**). Sequence alignment, structural superimposition, and previous biochemical data suggest that these domains all bind to CheY-like proteins [13] (**S1A–S1B Fig**), although whether different P2 domains are specific for one CheY isoform over another (i.e. $CheY_{1-3}$) is not currently known. In addition to the P2 domain, the other major difference between $CheA_1$ and $CheA_2$ is the length of the P3 domains (**Fig 1D**). $CheA_1$ has a P3 domain very similar in length to TmCheA, whereas in $CheA_2$ the P3 is over 68 residues and 43 Å longer than $CheA_1$ and TmCheA. Previously, it has been recognized that *B. burgdorferi* and *Treponema denticola* have extended P3 domain structures [21,24]. It was hypothesized that these extended P3 domains may stabilize the $CheA_2$ homodimer by increasing the binding interface surface area of the two CheA subunits. Another possibility could be that the longer P3-P3' four-helix bundle is an adaptation to allow better engagement with the chemosensory arrays of spirochetes, which have distinct architectures owing to the high membrane curvature [21,24]. Although the exact molecular mechanism behind why *Borreliaceae* $CheA_2$ encodes for a second P2 domain and an extended P3 domain has yet to be determined, it is nonetheless interesting that these two features are conserved across the family of *Borreliaceae* (**S3 Fig**).

Motility and chemotaxis contribute significantly to the pathogenesis of *B. burgdorferi* as loss of motility or chemotaxis resulted in non-infectious spirochetes that failed to colonize tick vectors and/or mammalian hosts [17,73,74]. A previous study from our group established that chemotaxis-signaling mediated by the $CheA_2$ histidine kinase is required for the transmission of spirochetes from ticks to mammals, as well as for establishing an infection in a vertebrate host, but is not essential for its survival in ticks [17]. Based on the organization of the chemotaxis gene clusters [75] as well as mutagenesis studies of genes in the $cheA_1$ [11,14–16] and $cheA_2$ [12,13,17,73] operons, we hypothesized that *B. burgdorferi* may utilize different chemosensory machineries in different environments during its infectious cycle, i.e., $CheA_2$ modulates chemotaxis during transmission/persistence in mammalian hosts while $CheA_1$ constitutes a second chemotaxis pathway to guide the chemotactic response in the tick vector and accordingly mediate *B. burgdorferi* transmission from infected ticks to mammals. In line with our hypothesis, a recent study by Iyer *et al*. showed that $CheA_1$ and $CheY_2$ are highly induced in transmitting ticks which indicates that these two proteins may form a functional chemotaxis pathway that is activated when an infected tick takes a blood meal so as to prepare dormant spirochetes for migration and transmission to a new host [72]. Lastly, a recent study by Caimano *et al*. further substantiated the role of $CheA_1$ during transmission whereby the level of $cheA_1$ is highly induced by RpoS while in fed ticks and mammals [76].

One interesting observation from the current study revealed that $CheA_1$ has a dynamic localization during growth. Importantly, the intense single pole localization of $CheA_1$ when cultivated under condition mimicking the unfed tick milieu versus elevated temperature indicates that $CheA_1$ may play a more prominent role in the arthropod vector as compared to the vertebrate host (**Figs 3** and **S6**). $CheA_1$ may be strategically placed near the cell pole in proximity to the chemoreceptor arrays [16], ready to detect stimuli from the blood meal in order to

initiate transmigration out from the tick midgut to the salivary glands for transmission [77,78]. To answer this question, *in vivo* visualization of $cheA_1^{mut}$ cells after the tick blood meal is necessary to reveal whether the mutant's inability to be transmitted via tick bite (**Table 4**) is due to failure in migration from the hemolymph to the tick salivary glands. Additionally, the increase in puncta localization of CheA$_1$-GFP as the cells transition from early to late log phase has a striking resemblance to ParB localization. ParB is a chromosome segregation protein that has been shown to recruit the structural maintenance of the chromosome (SMC) complex to the origin of replication in *B. burgdorferi* [79]. It would be interesting to test if the localization of CheA$_1$ is associated with the DNA partitioning Par proteins perhaps to ensure equal segregation of CheA$_1$ protein among the daughter cells during cell division.

Given that the $cheW_2$ operon is positively regulated by RpoS and BosR [18,76], the failure of $cheA_1^{mut}$ during transmission could be partly due to defect in transmigration from the tick gut to the salivary glands during the blood meal, similar to the phenotype of a *rpoS* mutant in the feeding tick [80]. Most significantly, the possible association of CheA$_1$ with the Par proteins implies that the established pool of CheA$_1$ needs to be distributed equally among the daughter cells. This unique partitioning was also observed in cells cultured at elevated temperature but only during the late stationary phase (**S6B Fig**). Once these late stationary phase cells were re-inoculated into fresh BSK-II media, CheA$_1$ localization reverted to the diffused pattern as seen during day 1. This protein localization is specific to CheA$_1$ as we did not observe this with CheA$_2$ (**S5D** and **S6C Fig**). The need for CheA$_1$ to have spatial and temporal localization is currently unknown. Given that the tick-mouse infection study points toward a role for CheA$_1$ during tick transmission and hematogenous dissemination, it would be interesting to monitor the localization of CheA$_1$ in feeding ticks and in mammals to determine if CheA$_1$ alters its localization in different cellular milieus and whether such localization contributes to its function.

Following initial deposition, either via tick bite or needle inoculation, *B. burgdorferi* needs to first establish a local infection before disseminating to peripheral organs. Therefore, hematogenous dissemination is a critical pathogenic event for early infection [81]. Once in the bloodstream, the spirochetes must transmigrate via vascular extravasation to spread and colonize distal organs [82,83]. Several findings elucidate the dissemination event in which CheA$_1$ is most likely to be involved. First, the $cheA_1$ mutant is able to survive at the injection site of immunecompetent and–deficient mice after needle inoculation, indicating that CheA$_1$ is not required for skin tissue survival. Needle inoculation into SCID mice allowed partial dissemination of the $cheA_1$ mutant to the bloodstream and subsequent colonization of skin regions (i.e., ears) distal from the injection site as well as other organs, but failed to do so in immunocompetent mice. This result indicates that CheA$_1$ is required for the spirochete survival in the bloodstream most likely by aiding the bacteria in immune evasion. In line with this, immunoblot analysis revealed that deletion of CheA$_1$ negatively impacted the expression of several essential surface virulence factors, such as OspC and BBK32 (**Fig 9B**). OspC has been shown to interact with host immune components, such as the complement component 4b (C4b), and contributes to the survival of *B. burgdorferi* in the bloodstream [42]. This can explain the finding in SCID mice where only 50% of the mice had systemic infection as the complement system remains intact in the immune-deficient mice and $cheA_1^{mut}$ is not as protected in the bloodstream as compared to the wild type (**Table 2** and **Fig 6**). This was corroborated by serum bactericidal assays, as deletion of CheA$_1$ renders the spirochetes more susceptible to serum killing (**Fig 7B**). *B. burgdorferi* infection by tail-vein further supported this proposition, as the mutant was cleared from the bloodstream of wild-type mice within 14 days and was severely attenuated in hematogenous dissemination. Tail-vein injection with high doses of spirochetes was not able to rescue the survival and systemic dissemination defect of $cheA_1$ mutant indicating that in addition to

attenuated innate and adaptive immune evasion, the mutant may have a defect in tissue colonization or an attenuated ability to extravasate due to dysregulated surface protein expression (**Fig 9**), such as BBK32 [84], which led to the clearance of spirochetes in the bloodstream by the host immune system. The ability of $cheA_1^{mut}$ to transmigrate from the vasculature will need to be examined via intravital imaging in order to determine if the reduced ability of $cheA_1^{mut}$ to colonize and disseminate from the bloodstream of mice is partially due to a defect in extravasation in addition to impaired immune evasion. Artificial infection of *Ixodes* ticks showed no significant variation in the survival between wild type and mutant (**Fig 8A**). Furthermore, the $cheA_1$ mutant can be acquired from the skin of infected mice suggesting that CheA$_1$ is dispensable for acquisition. In addition, $cheA_1^{mut}$ failed to be transmitted via infected tick bites, suggesting that CheA$_1$ is required for tick transmission, consistent with the observed increase in $cheA_1$ transcripts during transmission [72]. Immunoblot and qRT-PCR analyses (**Fig 9**) suggest that the attenuated infectivity in mice upon depletion of CheA$_1$ is likely due to dysregulated RpoS and RpoS-regulon expression, which are essential for *B. burgdorferi* survival, immune evasion, and persistence in mammalian hosts [42,43,45,69,78,85,86]. Protein turnover assays revealed that the stability of RpoS was significantly reduced in the absence of CheA$_1$ (**Fig 9D**) and that this is likely due to dysregulated protease expression (**Fig 10**).

In other bacteria such as *E. coli* and *Bacillus subtilis*, in addition to the $\sigma^{70}$ house keeping sigma factor, , multiple alternative sigma factors are employed to effectively regulate gene expression based on changes in the growth condition, such as stress response, starvation, and transition between motile and sessile stages [87,88]. The activity of each σ is controlled by the binding of anti-σ proteins such that the proper σ can be released for interacting with the RNA polymerase (RNAP) when required [89–91]. The genome of *B. burgdorferi* encodes two alternative sigma factors, $\sigma^{54}$ (RpoN) and $\sigma^{38}$ (RpoS) [9]. How sigma factor partner switching occurs in *B. burgdorferi* remains unclear as no anti-sigma factors have yet been discovered. Preliminary RNA-seq analysis performed using *in vitro* cultured samples revealed significant up-regulation of many RpoS-repressed genes [76], such as *bb_d18* and *bb_a68* [76,92] in the $cheA_1^{mut}$ mutant (**S7 Fig**). As shown in Fig 10A, the level of *bb_d18* was significantly higher in $cheA_1^{mut}$ as compared to WT when validated using qRT-PCR. Notably, the expression of *bb_d18* remained elevated in the complemented strain $cheA_1^{com}$, which may explain why only a partial complementation was achieved in $cheA_1^{com}$ (**Table 1**). The untimely expression of RpoS-repressed genes may have diluted the pool of RNAP-RpoD complex available for transcription of core genes, such as *clpX* and *lon* proteases, leading to global dysregulation of the CheA$_1$ mutant transcriptome (**S8 Fig**). Additionally, BBD18 is known as a negative regulator of RpoS [93–95] and upregulation of this gene can further repress the expression of RpoS-regulon in the mutant (**Fig 9**). In sum, the data strongly suggest that CheA$_1$ affects the pathogenicity and transcriptome of *B. burgdorferi* by influencing the stability of RpoS potentially through ClpX and Lon proteases.

As discussed above, CheA$_1$ may contribute to chemotaxis within the tick. It is also possible that it has a role to play in host dissemination. We did not detect any chemotaxis defect in the $cheA_1^{mut}$ *in vitro* (**Fig 4**); however, the significance of CheA$_1$ in chemotaxis may only be observable *in vivo* as lab cultivation rarely accurately represents any aspect of the natural infectious cycle of *B. burgdorferi* [96]. We had previously shown that CheW$_2$ contributes to the full pathogenicity of *B. burgdorferi* as depletion of CheW$_2$ resulted in attenuated dissemination and colonization of the spirochetes [18]. Given that CheW$_2$ preferentially interacts with CheA$_1$ [16], CheA$_1$, CheW$_2$, and CheY$_2$ may constitute an active chemotaxis pathway during the vector transmission and host dissemination phases to guide the chemotactic responses and migration of *B. burgdorferi*. Notably, the phenotype of $cheA_1^{mut}$ differs from what was observed in an infectious *cheY_2* and *cheW_2* mutant; Xu *et al.* showed that *cheY_2* mutant was cleared from

the inoculation site after two weeks and failed to cause systemic infection [15]. In contrast, $cheA_1{}^{mut}$ persisted at the initial injection site and was able to replicate as shown by our time course study (**Fig 5**). As for CheW$_2$, even though deletion of CheW$_2$ led to attenuated infectivity in immune-competent mice, its infectivity was not affected in SCID mice, which differs significantly from the *in vivo* phenotype of $cheA_1{}^{mut}$ [18]. However, all three mutants exhibited wild-type chemotaxis behavior *in vitro* which raises the question as to whether CheA$_1$-CheW$_2$-CheY$_2$ is required for chemotaxis under specific circumstances *in vivo*. *cheA$_1$* and *cheY$_2$* mutant survival in the tick vector was not significantly different from the wild type (**Fig 8A**), emphasizing that CheA$_1$ and CheY$_2$ [15] are not required for the persistence and survival of *B. burgdorferi* in the arthropod vector. How is $cheA_1{}^{mut}$ able to persist and spread in the skin tissue but not to distal organs? Given that *B. burgdorferi* can spread through direct tissue spreading [81], it is possible that once $cheA_1{}^{mut}$ establishes localized infection at the injection site, it is able to spread within the skin tissue to distal skin regions (e.g., ears) and avoid clearance by the host immune system via non-hematogenous spreading.

In order to colonize and survive in a mammalian host, *B. burgdorferi* has evolved to alter its surface antigen expression for distinct tissue tropism as well as in response to host immune reaction [45]. The disparity in host immune repones elicited by WT vs $cheA_1{}^{mut}$ (**Fig 11**) is in congruence with the mouse infection results. The lack of key inflammatory mediators necessary for spirochete dissemination may dampen the progression of $cheA_1{}^{mut}$ infection and dissemination outcome, leading to attenuated infectivity as observed (**Tables 1, 2 and 3**). The anti-inflammatory cytokine IL-10 plays a key role in the progression of *B. burgdorferi* infection [60–62]. The failure of $cheA_1{}^{mut}$ to elicit timely expression of IL-10 in mice may have contributed to its failure in dissemination and colonization in both immune-competent and–deficient mice (**Tables 1 and 2**). Despite this, the mutant was able to persist at the inoculation site (**Fig 5**) without the protection of an IL-10 cytokine effect which implies that additional bacterial and host factors may be required. Lack of an activated host immune reaction following $cheA_1{}^{mut}$ infection (**Fig 11B**) could potentially allow the spirochetes to evade immune surveillance and persist at the inoculation site.

In summary, this study reveals that CheA$_1$ is a non-canonical chemotaxis histidine kinase with a novel role in *B. burgdorferi* infectivity and control of virulence factor expression. As a histidine kinase, CheA$_1$ does not directly influence gene expression but must act through a response regulatory protein in order to activate a given signaling pathway. Identifying the cognate response regulator for CheA$_1$ is the next essential step in comprehending how CheA$_1$ influences global transcriptomic changes via RpoS protein expression and stability as well as the greater roles that CheA$_1$ plays in *B. burgdorferi* pathogenesis *in vivo*.

## Material and methods

### Ethics statement

All animal experimentation was conducted following the NIH guidelines for housing and care of laboratory animals and performed in accordance with the Virginia Commonwealth University institutional regulations after review and approval by the Institutional Animal Care and Use Committees [IACUC approval number: #AD10001779].

### Bacterial strains and growth conditions

Infectious clone A3-68 and A3-68Δ*bbe02* (wild type), two derivative strains from the *B. burgdorferi* sensu stricto B31-A3, were used in this study [25]. They were a kind gift from P. Rosa (Rocky Mountain Laboratories, NIAID, NIH). Cells were grown in Barbour-Stoenner-Kelly (BSK-II) medium as previously described [97] in appropriate antibiotic(s) for selective

pressure as needed: streptomycin (50 μg/ml), kanamycin (300 μg/ml), and/or gentamicin (40 μg/ml). To determine the expression of CheA$_1$, $10^5$ stationary phase wild type cells were inoculated into 10 ml of fresh BSK-II medium and then cultured at 23˚C/pH 7.6 (unfed tick, UF) or 34˚C/pH 7.6 (a routine laboratory cultural condition). To examine the expression of RpoS and RpoS-regulon, $10^5$ stationary phase wild type, mutants, and complemented strains were inoculated into 10 ml of fresh BSK-II medium and then cultivated at 37˚C/pH 6.8 with 5% $CO_2$ to mimic the host condition. Cells were harvested for immunoblot analysis upon entry into stationary phase at ~$10^8$ cells/ml.

### CheA$_1$, CheA$_2$, and TmCheA AlphaFold model generation and analysis

To build CheA$_1$, CheA$_2$, and TmCheA models, protein sequences for TmCheA (AAA96387.1), CheA$_1$ (AAC66935.1), and CheA$_2$ (AAC67024.1) were retrieved from the NCBI database and submitted for automated model-building using AlphaFold2 [20]. All parameters were kept at their default values for model building.

### Construction of CheA$_1$/pBBE22 for complementation of *cheA$_1$* mutant

pGA$_1$kan, a previously constructed vector with an internal 393-bp deletion in *cheA$_1$* gene (nucleotides 1,016–1,409) [11], was used to inactivate the *cheA$_1$* in B31 A3-68 and A3-68Δ*bbe02* strains via allelic exchange mutagenesis. All the experiments performed in this report was done using mutants generated in the A3-68Δ*bbe02* background except for RpoS protein turnover analysis in which the *cheA$_1$* mutant was generated in the B31 A3-68 background because A3-68Δ*bbe02* carries a streptomycin resistant cassette. To complement the *cheA$_1$* mutant, CheA$_1$/pBBE22 (**Fig 2A**), was constructed via PCR ligation using primer pair P$_1$/P$_2$ to amplify *bb0566-cheA$_1$* region and cloned into pBBE22G vector at BamHI restriction site [17,98,99]. All primers used in this study are listed in **Table 5**.

### Construction of CheA$_1$IFD for in-frame deletion of *cheA$_1$*

To create an alternative *cheA$_1$* mutant, an in-frame deletion construct was made using PCR ligation method to in-frame replace the entire open reading frame (*orf*) of *cheA$_1$* with kanamycin (*kan*) cassette. The left and right arm of the deletion construct was PCR amplified using P$_{27}$/P$_{28}$ and P$_{29}$/P$_{30}$, respectively, and the *kan* cassette was amplified using P$_{31}$/P$_{32}$ (**Table 5**). The resulting three amplicons were then PCR ligated using P$_{27}$/P$_{30}$ and ligated to pJET1.2/ blunt cloning vector (Thermo Scientific, Waltham, MA) yielding CheA$_1$IFD (**S4B Fig**).

### Localization of CheA$_1$ and CheA$_2$ in *B. burgdorferi*

For the localization of CheA$_1$ and CheA$_2$, *cheA$_1$-gfp*/pBSV2G (**S3A Fig**) and *cheA$_2$-gfp*/ pBSV2G (**S3B Fig**) constructs were used to complement *cheA$_1^{mut}$* and a previously constructed *cheA$_2$* mutant strain [11], respectively. A control plasmid with *gfp* driven by the *flgB* promoter was included as a control for the localization study [100]. To determine the localization of CheA$_1$ and CheA$_2$, GFP construct expressing cells were inoculated into 10 ml of fresh BSK-II medium and then cultured at 23˚C/pH 7.6 or 34˚C/pH 7.6. Cells were observed every 48 hours for up to 2 weeks. Images were taken using a Zeiss Axiostar plus microscope and processed using Axiovision software (Zeiss, Germany) as previously described [100].

### SDS-PAGE and immunoblots

10 to 20 μg of *B. burgdorferi* whole cell lysates were separated on 10 or 12% Stain-Free SDS-PAGE gel and transferred to PVDF membrane (Bio-Rad Laboratories, Hercules, CA).

**Table 5. Oligonucleotide primers used in this study.[e]**

| Primer | Description | Sequences |
|---|---|---|
| P$_1$ | complementation, *cheA₁* (F) | 5'– GGATCCCAGCAATAGGAGTTTTTAG –3' |
| P$_2$ | complementation, *cheA₁* (R) | 5'– GGATCCTTATTTTATAAGTTTAGTTATTGCA –3' |
| P$_3$ | qRT-PCR, mouse *β-actin* (F) | 5'– AGAGGGAAATCGTGCGTGAC–3' |
| P$_4$ | qRT-PCR, mouse *β-actin* (R) | 5'– CAATAGTGATGACCTGGCCGT–3' |
| P$_5$ | qRT-PCR, tick *β-actin* (F) | 5'– GATGACCCAGATCATGTTCG –3' |
| P$_6$ | qRT-PCR, tick *β-actin* (R) | 5'– GCCGATGGTGATCACCTG –3' |
| P$_7$ | qRT-PCR, *dnaK* (F) | 5'–CTCATGCGTAGCTATAATGGAGC–3' |
| P$_8$ | qRT-PCR, *dnaK* (R) | 5'–AAGAGTTGCTGCTGAGATCTC–3' |
| P$_9$ | qRT-PCR, *rpoS* (F) | 5'–ACCTATCTCCTGCTCAGTATATAA–3' |
| P$_{10}$ | qRT-PCR, *rpoS* (R) | 5'–CAAGGGTAATTTCAGGGTTAAAAG–3' |
| P$_{11}$ | qRT-PCR, *ospC* (F) | 5'–TGTTACTGATGCTGATGCAA–3' |
| P$_{12}$ | qRT-PCR, *ospC* (R) | 5'–AAGCTCTTTAACTGAATTAGC–3' |
| P$_{13}$ | qRT-PCR, *dbpA* (F) | 5'–GGACTAACAGGAGCAACA–3' |
| P$_{14}$ | qRT-PCR, *dbpA* (R) | 5'–CACCACTACTTCCAGTTTC–3' |
| P$_{15}$ | qRT-PCR, *ospA* (F) | 5'–GCAGCCTTGACGAGAAAAACAG–3' |
| P$_{16}$ | qRT-PCR, *ospA* (R) | 5'–CGCCTTCAAGTACTCCAGATCC–3' |
| P$_{17}$ | qRT-PCR, *lon-2* (F) | 5'–GTGGTACAGTTCTTCCTGTTGA–3' |
| P$_{18}$ | qRT-PCR, *lon-2* (R) | 5'–AGCTGTGCACTCTCTTTCATAA–3' |
| P$_{19}$ | qRT-PCR, *lon-1* (F) | 5'–GAGTTGAAGGGAGAGCCTTTAG–3' |
| P$_{20}$ | qRT-PCR, *lon-1* (R) | 5'–CGCCAAGGAAGCTCAGTAATA–3' |
| P$_{21}$ | qRT-PCR, *clpX* (F) | 5'–GTGTAAGCCGCTAGATTCTAAGTC–3' |
| P$_{22}$ | qRT-PCR, *clpX* (R) | 5'–GCCTGTAGGACCAACCAAAAG–3' |
| P$_{23}$ | *clpX* IFD, left arm (F) | 5'– TAGGTGTAAGTTATGAGGAG –3' |
| P$_{24}$ | *clpX* IFD, left arm (R) | 5'– ACGTTTCCCGTTGAATATGGCTCATAAAAACTTT TACCGAAAATAAAAC –3' |
| P$_{25}$ | *clpX* IFD, right arm (F) | 5'– TTTGATGCTCGATGAGTTTTTCTAAGTTGTTGTT ACAAAAGAATCTGT –3' |
| P$_{26}$ | *clpX* IFD, right arm (R) | 5'– ACAAGCTTACTTGGATTCTC –3' |
| P$_{27}$ | *cheA₁* IFD (in-frame deletion), left arm (F) | 5'– GTTGAGGTTTTAGAATATACTAAGATATC –3' |
| P$_{28}$ | *cheA₁* IFD, left arm (R) | 5'– GAATATGGCTCATAAGCTTTCCTTAAATC –3' |
| P$_{29}$ | *cheA₁* IFD, right arm (F) | 5'– CGATGAGTT TTTCTAAGTTGTTGACATAG ATGCAATAAC –3' |
| P$_{30}$ | *cheA₁* IFD, right arm (R) | 5'– CCTCCAGGCATATGCTGAAC –3' |

*(Continued)*

**Table 5.** (Continued)

| Primer | Description | Sequences |
|---|---|---|
| $P_{31}$ | *kan* cassette (F) | 5'– ATGAGCCATATTCAACGGGAAAC –3' |
| $P_{32}$ | *kan* cassette (R) | 5'– TTAGAAAAACTCATCGAGCATCAAATG –3' |
| $P_{33}$ | qRT-PCR, mouses *il-10* (F) | 5'– GCTCTTACTGACTGGCATGAG –3' |
| $P_{34}$ | qRT-PCR, mouses *il-10* (R) | 5'– CGCAGCTCTAGGAGCATGTG –3' |
| $P_{35}$ | qRT-PCR, mouses *il-1β* (F) | 5'– GAAATGCCACCTTTTGACAGTG–3' |
| $P_{36}$ | qRT-PCR, mouses *il-1β* (R) | 5'– TGGATGCTCTCATCAGGACAG –3' |
| $P_{37}$ | qRT-PCR, mouses *tnf* (F) | 5'– CTGAACTTCGGGGTGATCGG –3' |
| $P_{38}$ | qRT-PCR, mouses *tnf* (R) | 5'– GGCTTGTCACTCGAATTTTGAGA –3' |
| $P_{39}$ | qRT-PCR, mouses *ccl-2* (F) | 5'– TTAAAAACCTGGATCGGAACCAA –3' |
| $P_{40}$ | qRT-PCR, mouses *ccl-2* (R) | 5'– GCATTAGCTTCAGATTTACGGGT –3' |

[e] The underlined sequences are the engineered restriction cut sites for DNA cloning; F, forward; R, reverse.

Stain-Free SDS-PAGE was first activated and imaged using ChemiDoc MP Imaging System (Bio-Rad Laboratories) prior to membrane transferring to visualize total proteins on the gel. The immunoblots were probed with antibodies against *B. burgdorferi* CheA$_1$, CheA$_2$ [11], P66 [101], BBK32 [102], RpoS, OspC, and DbpA [86]. DnaK was used as an internal control, as previously described [97]. Membranes were developed using horseradish peroxidase secondary antibody with an ECL luminol assay or with fluorescently labelled secondary antibodies. Signals were imaged using the ChemiDoc MP Imaging System and quantified with the Image Lab software (Bio-Rad Laboratories).

## Bacterial motion tracking analysis, swimming plate, and capillary tube-based chemotaxis assays

The swimming velocity of *B. burgdorferi* cells was measured using a computer-based motion tracking system as previously described [31]. Swimming plate assays were conducted as previously documented [11,32,103]. The diameters of swimming rings were recorded in millimeters. Wild-type A3-68 *Δbbe02* was used as a positive control. A non-motile strain, *flgE* mutant, [103] was used as a negative control to monitor the initial inoculum size. For the capillary tube assay, *N*-acetyl-D-glucosamine (NAG) was used as an attractant as previously reported [31]. The spirochete cells accumulated in the capillary tubes were enumerated using Petroff-Hausser counting chambers. The mean of three replicates was determined and the data are expressed as the mean relative increase over a buffer control containing no attractant for each group. A two- or more fold increase in the number of spirochetes in comparison to buffer control is considered as significant.

## Quantitative reverse transcription PCR (qRT-PCR)

Ticks and mouse tissue samples for qRT-PCR analysis were prepared as described [17,97]. Briefly, total RNA from ticks or mouse tissues was isolated using TRIzol reagent (Invitrogen,

Carlsbad, CA) and contaminating genomic DNA was removed using Turbo DNase (Ambion, Austin, TX). The DNase-treated RNAs were re-purified and converted to cDNA using Super-Script IV VILO Master Mix (Invitrogen) according to the manufacturer's instructions. Quantitative PCR was then performed using Fast SYBR Green Master Mix (Applied Biosystems, Foster City, CA). The spirochete burdens within infected mice and ticks were expressed as the *flaB* transcript levels relative to the copy number of the mouse or tick *β-actin* transcript levels as described previously [17]. Primers used in qRT-PCR are summarized in **Table 5**. Data are expressed as the mean fold change of two replicates relative to WT ± SEM. The significance of the difference between different experimental groups was evaluated with ANOVA (*P* value < 0.05).

### Protein turnover assay

To determine the stability of RpoS protein, protein turnover was performed as described previously [104] using wild-type A3-68 and a $cheA_1$ mutant constructed in the same background. The mutant was confirmed to have the same antigen profile as $cheA_1{}^{mut}$ [103,104]. Briefly $10^5$ cells/ml of *B. burgdorferi* cells were inoculated into 50 ml of BSK-II medium, pH 6.8, and cultivated at 37˚C to mid log phase ($5x10^6$ cells/ml) or stationary phase ($10^8$ cells/ml). 100 µg/ml of spectinomycin was added to the culture to arrest protein synthesis. 5 ml of cells were harvested at 0, 1, 2, 3, 4, 6, 8, and 24 hours post-protein arrest for SDS-PAGE followed by immunoblots against RpoS and DnaK.

### Mouse infection studies

BALB/c or BALB/c SCID mice at 6–8 weeks of age (Jackson Laboratory, Bar Harbor, MN) were used in the needle infection study. The animal studies were carried out as previously described [17,97]. Briefly, mice were given a single subcutaneous injection of $10^5$ spirochetes and sacrificed 3 weeks post-infection. For time point inoculation site study, mice were similarly infected with WT or $cheA_1{}^{mut}$ and the initial inoculation sites were marked for harvesting at 96 hour, 14 day, 21 day, and 28 day post-infection. Skin specimens were subjected to qRT-PCR analysis as described above.

For tail-vein infection, $10^6$ spirochetes re-suspended in 100 µl PBS was injected into the tail-vein of BALB/c mice. The non-motile non-infectious *flaB* mutant was included as a negative control [32]. Mice were sacrificed two weeks after infection. Tissues from the ear, skin, joint, and heart were harvested and placed into 1 ml BSK-II medium containing 5 µg/ml of rifampicin. The samples were incubated at 34˚C for up to one month and microscopically monitored for the presence of spirochetes. Tissues were similarly collected for qRT-PCR analysis to assess for spirochete burdens. For bulk RNA-seq analysis, $10^7$ spirochetes in 100 µl of BSK-II was injected into the dorsal side of BALB/c mice at three separate locations. For sham control, equal volume of BSK-II medium was injected instead. Mice were sacrificed at 72-hour post-injection. Skin tissues from the injection sites were harvested for RNA extraction as described above.

### Serum bactericidal assays

Serum bactericidal assay was performed as previously described [43]. Normal human serum was purchased from BioIVT (Westbury, NY). (Briefly, wild-type A3-68Δ*bbe02*, $cheA_1{}^{mut}$ and $cheA_1{}^{com}$ were grown to mid log phase at 34˚C/pH 7.6. $10^6$ cells in 80 µl of BSK-II medium were inoculated into a 96-well plate followed by the addition of 20 µl of normal human serum (NHS) or heat inactivated NHS (hiNHS) to get a final concentration of 20% serum in 100 µl. Cells were incubated at 34˚C for 2 h. The percentage of viable cells after incubation were

enumerated under dark field microscope using Petroff-Hausser counting chambers based on immobilization, cell lysis, and loss of cell membrane integrity. Experiments were repeated at least three times. Data is presented as percent viable cells relative to input cells.

## Tick-mouse transmission study

The tick-mouse transmission study was carried out as previously described [17,41]. Briefly, naïve *Ixodes scapularis* nymph (Oklahoma State University, Stillwater, OK) were artificially infected by microinjection as previously described [39]. Two days after injection, infected nymphs were allowed to parasitize on naïve C3H mice (5 ticks/mouse and 3 mice for each *B. burgdorferi* strain) for 5 to 7 days and allowed to fall off. Engorged ticks were collected and subjected to qRT-PCR analysis to determine spirochete burdens. At day 14 after the tick feeding, mice were sacrificed and tissues from the skin, heart, joint, bladder and blood were harvested for quantification of spirochete burden via qRT-PCR analysis to determine transmission rate as described previously [41]. Spirochetal burden in fed ticks is presented as log copies of *flaB* / $10^3$ tick *β-actin* copies while transmission data is shown as *flaB* copies per $10^6$ mouse *β-actin*.

## Tick acquisition studies

To examine the ability of *cheA$_1$*$^{mut}$ to be acquired by feeding ticks, 1 x $10^4$ of either WT or *cheA$_1$*$^{mut}$ was injected subcutaneously into two naïve C3H mice each. Two weeks post-infection, skin biopsy was obtained from the site of injection to confirm present of live spirochetes via culturing. 2–3 naïve *I. scapularis* nymphs were placed onto the infected mice at the site of inoculation and allowed to feed to repletion. After 72 hours, fed nymphs were collected and tested individually for the presence of *flaB* via qPCR. Tick acquisition experiments were repeated twice and the mean of both data sets are presented as *flaB* copies per fed nymph.

## Construction of an in-frame deletion mutant of *clpX*

An in-frame deletion construct of *clpX* was made using the same PCR ligation method as described for CheA$_1$IFD. The *orf* of *clpX* was replaced with *kan* cassette. The left and right arm of the deletion construct was PCR amplified using P$_{23}$/P$_{24}$ and P$_{25}$/P$_{26}$, respectively and *kan* cassette was amplified using P$_{31}$/P$_{32}$ (**Table 5**). The resulting three amplicons were then PCR ligated using P$_{39}$/P$_{42}$ and ligated to pJET1.2/blunt cloning vector (Thermo Scientific) and used for transformation into wild-type A3-68 strain.

## Mouse RNA-seq analysis

Total RNAs were extracted from mouse skin tissues using TRIzol reagent as described and treated with DNase to remove contaminating genomic DNA. RNA samples were sent to CD Genomics (CD Genomics, Shirley, NY) for RNA-seq analysis. Briefly, rRNA was depleted using Illumina Ribo-Zero Plus rRNA Depletion Kit. Library preparation was performed with NEBNext Ultra II RNA (non-directional) Kit. Paired-end reads with a length of ~150 nt were generated with NovaSeq 6000 platform. Quality control was performed with Fastp with low-quality reads filtered. Reads were aligned to the host reference genome (GRCm39). Quantification was then performed based on the alignment results, followed by differentially expressed gene analyses with DESeq2. Significant differentially expressed genes (DEG) between pairwise comparisons were identified with threshold |log2FC| > 0.5, and $p < 0.01$. All the significant DEGs were annotated with eggNOG, COG, Pfam, Swiss-Port, and KEGG databases. GO enrichment analysis and KEGG pathway analysis were performed with ClusterProfiler in R.

### Statistical analysis

For the swimming plates, motion tracking, capillary tube assays, and the mouse and tick infection studies, the results are expressed as means ± standard error of the mean (SEM). The significance of the difference between different experimental groups was evaluated with an unpaired Student *t* test or ANOVA ($P$ value < 0.05). For serum bactericidal assay, statistical significance was determined using multiple *t* test ($P$ value < 0.05) followed by the Holm–Bonferroni method to correct for multiple comparisons. Data is depicted as the mean of all replicates ± SEM.

## Supporting information

**S1 Fig. Sequence and structural alignment of the P2 domains from BbCheA$_1$ and BbCheA$_2$.** (A) Sequence alignment of BbCheA$_2$ P2$^\alpha$ and P2$^\beta$ domains compared to TmCheA P2 domain. Conserved residues at the CheY:P2 domain binding site interface are indicated with black boxes. (B) Structural superimposition of BbCheA$_1$ P2 (pink) and BbCheA$_2$ P2$^\alpha$ (green) domains with TmCheA P2:CheY complex (tan, PDB:1U0S) [5]. Multiple sequence alignment analysis (MSA) were generated using Clustal Omega [6] and figures were prepared in PyMol [7].
(TIFF)

**S2 Fig. Multiple sequence alignment analysis (MSA) of *Treponema* spp. that contain a single CheA isoform.** (A) MSA of *Treponema* spp. CheA P2 domain sequences. BbCheA$_1$ and BbCheA$_2$ sequences are included for comparison. The location of BbCheA$_2$ P2$^\alpha$ is marked with a black box. (B) MSA of *Treponema* spp. CheA P3 domain sequences. BbCheA$_1$ and BbCheA$_2$ sequences are included for comparison. The location of BbCheA$_2$ P3 domain is marked with a black box. Sequences collected using Annotree [1], MSA files generated using Clustal Omega [6].
(TIFF)

**S3 Fig. Multiple sequence alignment analysis (MSA) of *Borrelia* spp. and *Borreliella* spp. CheA$_1$ and CheA$_2$ sequences.** (A) MSA of *Borrelia* (B_) and *Borreliella* (Ba_) spp. CheA$_1$ and CheA$_2$ P2 domain sequences. Location of BbCheA P2$^\alpha$ marked with black box. (B) MSA of *Borrelia* (B_) and *Borreliella* (Ba_) spp. CheA$_1$ and CheA$_2$ P3 domain sequences. Location of BbCheA$_2$ extended P3 domain sequence marking with black box. Sequences collected using Annotree [1], MSA files generated using Clustal Omega [6].
(TIFF)

**S4 Fig. Detection of plasmid content in the *cheA1com* strain by PCR and construction of CheA$_1$IFD for the in-frame deletion of *cheA$_1$* gene.** PCR was used to detect the plasmid profile of WT (A), *cheA$_1$$^{mut}$* (B) and *cheA$_1$$^{com}$* (C). The primers used were described previously [8]. (D) A diagram illustrating the construction of CheA$_1$IFD. To construct an in frame deletion mutant, primer pair P$_{27}$/P$_{28}$ and P$_{29}$/P$_{30}$ were used to amplify the upstream and downstream flanking region of *cheA$_1$*. Primer pair P$_{31}$/P$_{32}$ was used to amplify a promoterless kanamycin cassette (*kan*). *cheA$_1$* was in-frame replaced by *kan* via PCR fusion technique with primer pair P$_{27}$/P$_{30}$. The resulting PCR fusion amplicon was cloned into pJet1.2 vector forming CheA$_1$IFD.
(TIFF)

**S5 Fig. Construction of *cheA$_1$-gfp*/pBSV2G and *cheA$_2$-gfp*/pBSV2G and the expression of CheA GFP fusion proteins.** (A) Construction of cheA1-gfp/pBSV2G. The upstream flanking region (*PcheW$_2$*, 417 bp) of *cheW$_2$* gene was PCR amplified and fused to *cheA$_1$* using the

engineered restriction sites as shown. The fused $PcheW_2$-$cheA_1$ fragment was then fused to $gfp$ gene containing a 5 x Gly linker at the indicated restriction sites and cloned into pBSV2G, a shuttle vector of *B. burgdorferi* [3], yielding $cheA_1$-$gfp$/pBSV2G. **(B)** Construction of $cheA_2$-$gfp$/pBSV6G. Similarly, the $flgB$ promoter [4] was PCR amplified and fused to $cheA_2$ gene followed by fusion to $gfp$ with 5 x Gly linker at the indicated restriction sites prior to cloning into pBSV2G. The obtained construct was used to complement a previously constructed $cheA_2$ mutant strain [9]. **(C)** Immunoblot analysis of GFP fusion proteins. Cell lysates from *B. burgdorferi* strains expressing GFP, $CheA_1$-GFP, or $CheA_2$-GFP cultured at UF conditions (23˚C/pH 7.6) or routine laboratory cultural condition (34˚C/pH 7.6) were analyzed on SDS-PAGE and probed with antibodies against GFP or DnaK (as loading control). No excessive degradation of GFP fusion proteins was observed under both culture conditions. **(D)** $CheA_2$-GFP does not have a polar localization like $CheA_1$-GFP. *B. burgdorferi* strain carrying $CheA_2$-GFP construct was cultivated under UF tick condition. $CheA_2$-GFP appeared diffused with no specific cellular localization observed.
(TIFF)

**S6 Fig. $CheA_1$-GFP localization is spatially- and temporally-regulated.** $10^5$ cells/ml of *B. burgdorferi* strains that express **(A)** GFP, **(B)** $CheA_1$-GFP, or **(C)** $CheA_2$-GFP were inoculated into 10 ml fresh BSK-II medium and cultivated at 34˚C/pH 7.6. Images were taken every two days at ×200 magnification using a Zeiss Axiostar Plus microscope. Scale bars represent 10 μm.
(TIFF)

**S7 Fig. Complementation of $CheA_1$ restored RpoS protein stability in $cheA_1{}^{com}$.** The stability of RpoS protein was examined in stationary phase of B31 A3-68 $cheA_1{}^{com}$ upon protein synthesis arrest with spectinomycin. Samples were harvested at the indicated time points and probed using antibody against RpoS and DnaK (as a loading control).
(TIFF)

**S8 Fig. Volcano plot of differentially expressed genes (DEG) between WT and $cheA_1{}^{mut}$.** A total of 1391 DEG between WT and mutant were plotted. Three down-regulated protease genes (*bb0613*, *bb0612*, and *bb0253*) and six up-regulated RpoS-repressed genes (*bb_a38*, *bb_a68*, *bb_d18*, *bb_i16*, *bb_i39*, *bb_j09*) in $cheA_1{}^{mut}$ were highlighted in the volcano plot.
(TIFF)

**S1 Table. Oligonucleotide primers used in this study.**
(PDF)

**S2 Table. *B. burgdorferi* RNA-seq analysis.**
(XLSX)

**S3 Table. Mouse RNA-seq analysis.**
(XLSX)

## Acknowledgments

Special thank gives to Dr. Utpal Pal for his assistance with tick microinjection study. We thank Eamen Ho and Weigang Qiu of Hunter College of City University of New York for visualization of RNA-Seq data. We also thank Dr. Patricia Rosa for providing *B. burgdorferi* A3-68 and A3-68Δ*bbe02* strains, Dr. Jennifer Coburn for providing P66 antibody, Dr. Melissa Caimano for providing OspC and DbpA antibody, Dr. Janakiram Seshu for providing BBK32 antibody, and Dr. Zhiming Ouyang for providing the *lon-1* mutant.

## Author Contributions

**Conceptualization:** Ching Wooen Sze, Brian R. Crane, Ira Schwartz, Chunhao Li.

**Data curation:** Ching Wooen Sze, Michael J. Lynch, Chunhao Li.

**Formal analysis:** Ching Wooen Sze, Michael J. Lynch, Radha Iyer, Chunhao Li.

**Funding acquisition:** Brian R. Crane, Chunhao Li.

**Investigation:** Ching Wooen Sze, Kai Zhang, Michael J. Lynch, Radha Iyer, Ira Schwartz, Chunhao Li.

**Methodology:** Ching Wooen Sze, Kai Zhang, Michael J. Lynch, Radha Iyer, Ira Schwartz, Chunhao Li.

**Project administration:** Ira Schwartz, Chunhao Li.

**Resources:** Brian R. Crane, Ira Schwartz, Chunhao Li.

**Supervision:** Brian R. Crane, Ira Schwartz, Chunhao Li.

**Validation:** Ching Wooen Sze, Kai Zhang, Michael J. Lynch, Brian R. Crane, Chunhao Li.

**Visualization:** Ching Wooen Sze, Michael J. Lynch, Chunhao Li.

**Writing – original draft:** Ching Wooen Sze, Michael J. Lynch, Ira Schwartz, Chunhao Li.

**Writing – review & editing:** Ching Wooen Sze, Michael J. Lynch, Brian R. Crane, Ira Schwartz, Chunhao Li.

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
