## [Decision Letter · Decision Letter 0]

8 Sep 2023

Chris,

Thank you very much for submitting your manuscript "A chemosensory-like histidine kinase is dispensable for chemotaxis but regulates the virulence of Borrelia burgdorferi through modulating the stability of RpoS" for consideration at PLOS Pathogens. Your manuscript was reviewed by members of the editorial board and by three independent reviewers. The reviewers and I are enthusiastic about your innovative and impressive study discovering a new facet of gene regulation in the Lyme disease spirochete. Considering the reviews (below), we would like to invite the resubmission of a revised version that addresses all the reviewers' comments. The reviewers have concerns about the interpretation and analyses of some of your data, especially the transcriptome experiment, as well as a considerable number of comments about the writing and presentation, which require your attention. Based on the reviews, we are likely to accept this manuscript for publication, providing that you modify the manuscript according to the review recommendations.

Sincerely,

Scott

D. Scott Samuels

Academic Editor

PLOS Pathogens

Nina Salama

Section Editor

PLOS Pathogens

Kasturi Haldar

Editor-in-Chief

PLOS Pathogens

orcid.org/0000-0001-5065-158X

Michael Malim

Editor-in-Chief

PLOS Pathogens

orcid.org/0000-0002-7699-2064

Academic Editor Comment:

There is a more recent review on gene regulation (with an extensive section on RpoS) than references 17, 29 and 105: Curr. Issues Mol. Biol. 2021, 42(1), 223-266; https://doi.org/10.21775/cimb.042.223

Reviewer Comments (if any, and for reference):

Reviewer's Responses to Questions

**Part I - Summary**

Reviewer #1: To maintain its enzootic cycle, Borrelia burgdorferi, the Lyme disease spirochete must employ directed motility and modulate its transcriptome to adapt to the markedly different arthropod and mammalian host environments. Historically, these two facets of Lyme disease research have been pursued along largely separate tracks. However, it has long been recognized that the spirochete must possess mechanisms for coordinating and integrating them. This complex, engaging manuscript by Sze et al. represents a major step towards bringing these two fundamental lines of investigation into alignment. Herein, they use a diverse and impressive technical battery to dissect the function of CheA1, one of the spirochete’s two chemotaxis-related histidine kinases, and compare it to CheA2, the spirochete's other chemotaxis-related histidine kinase. They elegantly demonstrate that CheA1 and CheA2 clearly differ with respect to predicted structures and cellular locations. In contrast to CheA2, which is clearly involved in motility and chemotaxis under in vitro conditions, they show that loss of CheA1 does not affect motility in vitro. However, similar to what they have reported previously for CheY2, the putative cognate response regulator for CheA1, they also show that loss of CheA1 prevents tick transmission. Surprisingly, they found that needle inoculated spirochetes lacking CheA1 remain confined to the injection site, a phenotype which they relate to diminished expression of RpoS-dependent virulence factors caused by dysregulated proteolytic degradation of the alternative sigma factor. Lastly, RNAseq analysis of the injection site revealed that spirochetes lacking CheA1 elicit a markedly decreased local inflammatory response. This is a novel, important, and exceptionally well executed study. Nevertheless, some aspects of the presentation need clarification to ensure that readers can fully appreciate the extraordinary findings being reported:

1. CheA1, chemotaxis, tick transmission, and dissemination in the mouse. Throughout the manuscript the authors maintain that CheA1 is critical for tick transmission. Since this almost certainly involves chemotaxis, it is not clear why they claim, based solely on in vitro data, that CheA1 is not required for chemotaxis. As they note towards the end of the Discussion in their 2017 CheY2 paper, one cannot consider in vitro studies to be definitive with respect to chemotaxis in vivo. Moreover, given that CheY2 is believed to be the cognate response regulator for CheA1, a closer comparison of the CheA1 and CheY2 phenotypes (which seem quite similar, length of duration at the injection site not withstanding) seems worthwhile. In both cases, this reviewer suggests describing CheA1 and CheY2 as dispensable for chemotaxis in vitro. I note parenthetically that the statement on p25 that CheA1 has no known effector seems puzzling given published work with CheY2. The authors appear to believe that the dissemination and survival defect of the cheA1 mutant in mice is due to diminished expression of RpoS-dependent gene products. The data, as presented, however, do not eliminate the possibility that CheA1, which is known to be RpoS-dependent, is involved in chemotaxis within the mammal.

2. The comparison of CheA1 and CheA2 structural models with the solved structure of Thermotoga maritime CheA is very interesting and leads to some testable hypotheses to explain functional differences between CheA1 and CheA2. However, the rationale provided for this first section of the paper – that CheA1 is dispensable for chemotaxis – is perplexing for two reasons. One is that, per above, the dispensability appears to be for chemotaxis in vitro. The second is that no data for CheA1 have been presented yet in the manuscript or published previously to provide context for the statement. The greater structural similarity between CheA1 and other bacterial CheAs further implies some involvement of CheA in chemotaxis even though it is less ‘spirochete-like’.

3. The authors present data that CheA1 and CheA2 are expressed constitutively under the in vitro conditions studied herein. However, they also note and seem to rely upon previous microarray studies by Iyer et al. showing that CheA1 and CheA2 are differentially expressed with divergent expression profiles in ticks and mammals.

4. Discussion page 25. The evidence presented shows that CheA1 controls expression of the alternative sigma factor RpoS at the protein level through a previously unrecognized role regulating transcription of the Clp proteolytic cascade. This unanticipated finding results from RNAseq analysis showing global transcriptional dysregulation in cheA1 mutant spirochetes. If there is less RpoS, it stands to reason that there would be more RNA polymerase with sigma70 with the resultant global effects. Consequently, the need to invoke phosphorylation of anti-sigma factors and sigma factor replacement is not apparent, not to mention the fact that, to the best of this reviewer’s knowledge, anti-sigma factors have not been described in B. burgdorferi (the proteolytic pathway for degradation of RpoS is not an anti-sigma factor pathway). Admittedly, tying all these complex pathways together mechanistically presents quite a challenge. In this reviewer’s opinion, the authors shouldn’t feel compelled to mechanistically explain all of their observations.

5. The authors are to be commended for going outside their comfort zone to characterize host transcriptional responses at sites inoculated with the cheA1 mutant. That the mutant gives rise to a global dampening of the local inflammatory response is another unexpected finding. However, the comments about IL-10 knockout mice at the end of the discussion are not exactly on point. IL-10 is an anti-inflammatory cytokine. In IL-10 knockouts, the inflammatory response is heightened, not diminished as shown here.

Reviewer #2: This study conducted a thorough exploration of the role of the atypical chemotaxis histidine kinase CheA1 in Borrelia burgdorferi. The study's robust approach, which combined protein structural modeling, subcellular localization studies, genetic mutant and complement analysis, tick and mice infection experiments, gene expression profiling, and comparative RNA-seq, has yielded a comprehensive understanding of the intricate mechanisms by which CheA1 influences RpoS stability and the overall survival strategy of B. burgdorferi within the blood. The paper is well written, and the conclusion is well supported by solid data.

Minor concern:

1. L239-251, chemotaxis experiments. Are these experiments performed at 23 ℃ or 34 ℃? Given that CheA1 showed a unique localization at 23 ℃, these experiments should be done under same conditions. Does CheA1-CheY2 crosstalk with CheA2-CheY3? Or is it possible that CheA1 phosphorylates CheY3 and then contributes to chemotaxis at a different condition?

2. L253-272: Animal infection experiments. Only 1 out of 5 mice was positive in the injection skin site for the complemented strain. This partial complementation was also observed in distal tissues and in tick-mice transmission (Fig. 8B). What is the reason for this partial complementation? Does cheA1 co-transcribe with other genes, so that expression of neighboring genes was affected in the mutant? Has the IFD mutant been complemented and that complement showed similar or different phenotypes? Was the mutant recovered from blood?

3. Mice RNA-seq (L775-786), Fig. 11; Fig. S8, RNA-seq. Has qRT-PCR been performed to confirm the RNA-seq data? Please add data statistical significance.

4. BBD18 has been reported to repress RpoS translation. In Fig. S8, BBD18 expression was shown to be increased in cheA1 mutant? These data should be confirmed by qRT-PCR and/or immunoblot in the mutant and complement.

Editorial suggestions:

1. L31-33 is awkward. Please revise it.

2. L103, L105, L205, L344, L377: change “is” to “was”

3. L120: revise the sentence “why CheA1 is dispensable for chemotaxis while CheA2 is required”, as it is heretofore unknow whether CheA1 is required for chemotaxis. The role of CheA1 in chemotaxis was described later in this manuscript.

4. L204: protein cellular localization. Does addition of a GFP tag at the C-terminus of CheA1 affect protein localization and function?

5. L184: change “contain” to “contained”; L185: change “does” to “did”

6. L183-184, Fig S4: Please add the plasmid profiles of the wild-type strain and the mutant

7. L188: Has gene transcription been examined under these conditions?

8. L336-348, Fig. 7B, it would be more informative to present the data as ratio of viable cells treated with NHS or hiNHS to the input (106 cells).

9. L360-374, tick-mice transmission: were spirochetes re-isolated from mice tissues like the experiments done in Table 1?

10. L395. Did CheA1 mutation affect the growth under this condition (37 ℃/pH6.8)?

11. L401-404, Fig. 9B. How many times have these experiments been repeated? Data for Fig. 9C, please add data statistical analysis.

12. L423-426, Fig. 10A. Data lack appropriate statistical analysis.

13. L442-443. L734-736. Why the animal infection route and the bacterial dose are different from other animal experiments in this study?

14. Revise the Y-axis of Fig. 11B.

15. L541. Change “finding” to “findings”.

16. L634-636. The authors propose that CheA1 may regulate expression through a response regulator. This can be discussed in more detail.

17. L712. “The mutant was confirmed to have the same antigen profile as cheA1 mut”. What does this mean?

18. L754-755, tick-mice transmission: “At day 7 after the tick feeding, mice were sacrificed”; L1189, “At day 14 after tick feeding, mice were sacrificed”. Please revise.

19. L768-773: CheA1IFD mutant was mentioned here. The creation of this mutant I the SI can be moved to L658-663.

20. Fig. 4, Fig. 5, please add data statistical analysis.

21. Fig. 9. Three independent experiments are assumed to be performed. In L1209-1210, qRT-PCR data, “Data from two replicates are expressed as mean fold change relative to WT ± SEM. *, significant difference (P < 0.05).” Please revise.

22. Fig. 10. Are these data drawn from two independent experiments as indicated in L1219-1220?

23. Fig S5B. please add data statistical analysis.

24. Given that CheW2 interacts with CheA1, the phenotype of the cheW2 mutant should be included in the discussion.

25. It will be very helpful to have a Strains and plasmids list Table or at least indicate the strains used in the Figure legends (e.g., tick-mice infection experiments), as the cheA1 mutation has been created using two different plasmids and in two different strains.

Reviewer #3: This is an impressive and thorough study by Sze et al., to examine the role of CheA1, a chemotaxis family histidine kinase, in the pathogenesis of B. burgdorferi. CheA1 does not appear to be involved in chemotaxis, under any in vitro investigated conditions, but plays an important role in establishing host infection as well as transmission from infected ticks. CheA1 regulates RpoS and its associated regulon, thus identifying a new regulator of this crucial, central pathway regulating host infectivity. CheA1 localization is temperature and growth phase regulated, displaying an intriuging and unusual puncta and polar localization suggesting localized specific function(s). The experiements are well designed and hypotheses extensively tested, especially in animal models making this study a robust examination of CheA1 function. Some experimental protocols need more explanation and some data should be analyzed slightly differently, but overall I strongly support publication following the suggested modifications.

**Part II – Major Issues: Key Experiments Required for Acceptance**

Reviewer #1: (No Response)

Reviewer #2: None

Reviewer #3: Fig. 11: You should compare the data from WT-infected to cheA1 mutant infected, instead of both separately to mock infected. Some clarification on what is being represented in Fig. 11A, that is, are these KEGG pathways upreg, downreg or some up and some down.

**Part III – Minor Issues: Editorial and Data Presentation Modifications**

Reviewer #1: While the data are strongly suggestive that CheA1 (and for that matter CheY2) are involved in tick transmission, one can argue that a definitive statement to this effect requires determining if spirochetes reach the tick feeding site. The authors are encouraged to perform this experiment.

Reviewer #2: Please see the summary

Reviewer #3: Line “genes” at the end of the sentence.

Lines 110-111 – “,” around “as well as our group”

Line 113 – comma after CheA1

Line 116 – omit the second “the”

Line 122 – I think you mean proteins not promoters

I would omit the paragraph from 154 to 168, except for the part about Borreliaceae (162 – 166) as it distracts from the focus on Borrelia.

Line 172- strains, not strain. Please describe pGA1kan at least briefly. Which strain is being used in fig 2, bbe02 KO or the other?

Lines 177 “immunoblots using anti-CheA1 antibodies and anti-DnaK antibodies as a control.”

Line 190 – “chemotaxis-related”

Line 192 “are” not “is”

Omit or move to the discussion sentences in lines 191 -195.

Line 198 – what are “quantitative immunoblots”? I don’t see any description in the materials and methods. I don’t think you need to make claims that might raise concerns about the validity of immunoblot quantitation, just state the levels of CheA1 didn’t change.

Line 219-220 – repeat, omit.

I am not sure what you mean by “temporally-regulate” lines 204 and 232. I think growth phase and temperature-regulated should be considered, in addition to spatially. Seems like there is complex regulation of CheA1 localization.

Line 244 – “wild-type”

Lines 246-247 – Are your measurements (error) really precise to 0.1 nm? I think you need to adjust your significant figures.

Lines 253-255 – Put the last part of the sentence first “A previously developed…was used to determine if CheA1….”

In the legend for table 1 please define CheA1IFD

Lines 276 and 318 – “105 cells” and line 277 “were”

Lines 278 and 294 – did you use qRT-PCR and not qPCR to quantify Bb loads in tissues?

Line 329 – “The tail-vein…

Line 349 – I would say “is likely” instead of “can be”, as you have used human serum for in vitro survival assays and the in vivo studies were done in mice.

Lines 353 – “strains”

Line 355 – I think you mean qPCR, not qRT-PCR.

Line 356-358 – Just say that it was not statistically different from WT or comp, as what you have shown is that it actually is not slightly lower.

Line 370 – I would not call it an artificial mouse infection, but rather needle inoculation of mice.

I have concerns about figure 8B. It seems like the cheA1com strain does not really complement the phenotype. How do you get error bars for the comp strain in the skin and heart if only one mouse had detectable flaB transcripts? Instead of quantification of flaB just present what tissues had detectable flaB transcripts, quantification doesn’t seem appropriate.

Line 383 – “the flaB gene”

Lines 385-388 – This sentence seems to contradict what was stated above and I’m not sure what this means. Also, the data in Fig. 8C seem suspect. I can’t tell how many nymphs were analyzed for the CheA1 mutant but it doesn’t seem possible that the mean could be where it is graphed based on all the data points. Also, the SEM must be larger than represented as numerous nymphs seem to have zero flaB copies.

Lines 402 and 407-408 – you state that the level of rpoS transcript was not significantly reduced but Fig. 9C shows that it is significant. Please correct the text.

Please explain how the experiment in Fig. 9D was done in the results section (i.e. cell density, how translation arrested). Also, if protein synthesis was arrested, how does the level of DnaK increase a lot? I am not sure you have arrested protein synthesis. Also, why was the CheA1 comp not done?

Line 428 – omit “are”

Line 429 – “suggest”, data is plural.

Lines 442-443 – 107 cells …were.

Line 460 – Add “Hk2” to the Rrp2-RpoN…

Line 463 – profiles

Line 497 – operons

Line 530 – the localization of CheA1 is not unparalleled as you just pointed out that ParB also has a similar localization, so I would omit this.

Line 541 – findings

Line 570 – mice not mouse

Line 638 – omit chemotaxis as you have shown CheA1 does not function in chemotaxis (at least in your experiments) and just say “pathogenesis in vivo.”

Line 743 – what is the source of your “normal human serum”?

In Fig. S7 why do so many Bb seem to not express GFP whether by itself or added to CheA1 or CheA2 (A-C)?

I have a few issues with Fig. 3B legend: Time-lapse implies the same cells were imaged. Use “time course”. It does seem to appear that the puncta are increasing, but not in all cells and quantification might be important. Finally, how do you know the puncta coincide with zones of new peptidoglycan formation?

Fig. 4. Line 1153 – I don’t see any numbers for the measurement of diameter of the rings. It would be good to include these data.

Were statistical analyses done for Figs. 4 and 5?

Please explain cells were determined to be alive or dead in the serum assay. If a cell is non-motile but no obvious morphological change is it considered to be viable or not?

It would be nice if the murine transcripts DE (Fig. 11) when comparing WT to CheA1 mut by a new volcano plot were verified by qRT-PCR, but not necessary.

PLOS authors have the option to publish the peer review history of their article (what does this mean?). If published, this will include your full peer review and any attached files.

Reviewer #1: No

Reviewer #2: No

Reviewer #3: No

Figure Files:

Data Requirements:

Reproducibility:

References:

---

## [Editor Report · Decision Letter 1]

14 Oct 2023

Chris,

We are pleased to inform you that your manuscript 'A chemosensory-like histidine kinase is dispensable for chemotaxis in vitro but regulates the virulence of Borrelia burgdorferi through modulating the stability of RpoS' has been provisionally accepted for publication in PLOS Pathogens.

Best wishes,

Scott

D. Scott Samuels

Academic Editor

PLOS Pathogens

Nina Salama

Section Editor

PLOS Pathogens

Kasturi Haldar

Editor-in-Chief

PLOS Pathogens

orcid.org/0000-0001-5065-158X

Michael Malim

Editor-in-Chief

PLOS Pathogens

orcid.org/0000-0002-7699-2064
---

## [Editor Report · Acceptance letter]

17 Nov 2023

Dear Professor Li,

We are delighted to inform you that your manuscript, "A chemosensory-like histidine kinase is dispensable for chemotaxis in vitro but regulates the virulence of Borrelia burgdorferi through modulating the stability of RpoS," has been formally accepted for publication in PLOS Pathogens.

Best regards,

Kasturi Haldar

Editor-in-Chief

PLOS Pathogens

orcid.org/0000-0001-5065-158X

Michael Malim

Editor-in-Chief

PLOS Pathogens

orcid.org/0000-0002-7699-2064